 # CRoW: Benchmarking Commonsense Reasoning in Real-World Tasks

**Mete Ismayilzada**    **Debjit Paul**[*]    **Syrielle Montariol**[*]    **Mor Geva**[◇]    **Antoine Bosselut**

EPFL, Switzerland

[◇] Google DeepMind

mahammad.ismayilzada@epfl.ch

## Abstract

Recent efforts in natural language processing (NLP) commonsense reasoning research have yielded a considerable number of new datasets and benchmarks. However, most of these datasets formulate commonsense reasoning challenges in artificial scenarios that are not reflective of the tasks which real-world NLP systems are designed to solve. In this work, we present CRoW, a manually-curated, multi-task benchmark that evaluates the ability of models to apply commonsense reasoning in the context of six real-world NLP tasks. CRoW is constructed using a multi-stage data collection pipeline that rewrites examples from existing datasets using commonsense-violating perturbations. We use CRoW to study how NLP systems perform across different dimensions of commonsense knowledge, such as physical, temporal, and social reasoning. We find a significant performance gap when NLP systems are evaluated on CRoW compared to humans, showcasing that commonsense reasoning is far from being solved in real-world task settings. We make our dataset and leaderboard available to the research community.[1]

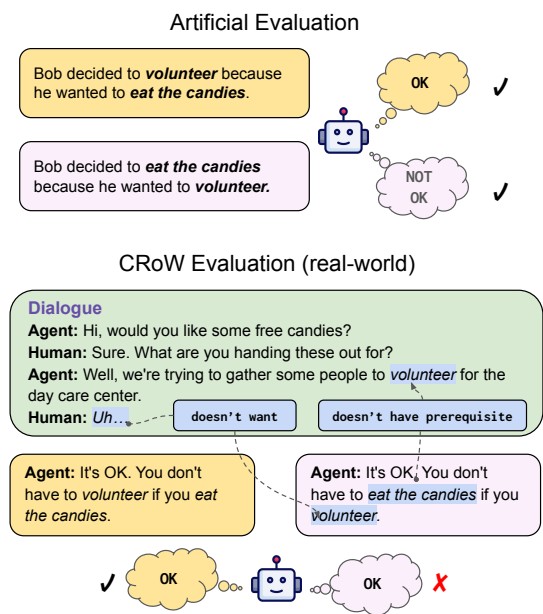

Figure 1: An example from one of the tasks (Dialogue) in our benchmark showcasing the difference between the evaluation of commonsense reasoning in an artificial and real-world setting. CRoW grounds this evaluation in a real-world context that often requires the use of rich and *implicit* commonsense knowledge to solve a task.

## 1 Introduction

Commonsense reasoning is a long-standing challenge in artificial intelligence (AI) and natural language processing (McCarthy, 1960; Winograd, 1974; Davis and Marcus, 2015; Choi, 2022), resulting in a large number of datasets and benchmarks designed to evaluate how AI systems reason in commonsense scenarios described in natural language (Davis, 2023). Recently, large language models, such as GPT-3 (Brown et al., 2020) and PaLM (Chowdhery et al., 2022), have demonstrated near-human performance on many of these benchmarks (Lourie et al., 2021). However, these models can still be brittle in practical deployments, raising

questions about how reliably these commonsense benchmarks truly evaluate the commonsense reasoning abilities of models.

Part of this issue stems from the practice that most commonsense datasets are designed to evaluate reasoning in artificial task settings that are not reflective of the real-world use cases in which NLP systems are deployed. In real-world settings, one almost never directly observes a test of commonsense knowledge in isolation. In this paper, we argue instead that *commonsense reasoning benchmarks should evaluate commonsense reasoning in the tasks in which these abilities are required.*

The necessity of commonsense to solve real-world tasks has been extensively argued since the

---

[*]Equal contribution

[1]https://github.com/mismayil/crow

early stages of AI, notably by Bar-Hillel (1960) in the context of machine translation. However, despite these early arguments, only recently was there an attempt to construct a commonsense reasoning dataset for machine translation (He et al., 2020), an effort which concluded that the commonsense reasoning abilities of modern models were still in their infancy when applied in real NLP tasks.

In this work, we build on these original ideas and introduce **CRoW**: a **C**ommonsense **R**easoning Benchmark for Real-**W**orld Tasks, a multi-task benchmark containing high-quality datasets for six real-world NLP tasks: machine translation (MT), open-domain dialogue (DG), dialogue summarization (DS), intent detection (ID), stance classification (SC), and safety detection (SD). Inspired by Winograd schemas (Levesque et al., 2011), we build our benchmark by applying commonsense-based minimal perturbations on examples from existing datasets for each task. For each of these tasks, we crowdsource collections of potential target references for the task, each grounded to a particular commonsense violation with respect to the original context (see Figure 1 for examples in dialogue response generation). We categorize these commonsense violations across six dimensions — temporal, causal, attribution, comparison, physical, and social — ensuring a diverse breakdown of commonsense reasoning types in CRoW.

Our empirical study across 13 state-of-the-art (SoTA) systems (including GPT-4) shows that CRoW is a challenging commonsense reasoning testbed, with the highest performing model scoring ∼18% lower than humans on individual examples and ∼37% lower on our more restrictive metric that evaluates situational robustness. Consequently, we provide CRoW to the community as the first commonsense benchmark specifically formed to test commonsense knowledge and reasoning abilities in the same contexts as real-world deployments of NLP systems. The contributions of our work can be summarized as follows:

- We design a common multi-stage data collection pipeline for generating commonsense-based Winograd-style variations of examples, which can be applied to many tasks.

- We apply our data collection pipeline to construct CRoW, a multi-task benchmark that evaluates the commonsense reasoning ability of models in solving six diverse real-world NLP tasks.

- For each task, we evaluate and analyze the perfor-

mance of state-of-the-art models on our benchmark across different dimensions of commonsense knowledge.

## 2  Related Work

**Commonsense Reasoning Benchmarks**   Many benchmarks measuring the commonsense reasoning abilities of state-of-the-art models have been released in recent years. Starting with the well-known Winograd Schema Challenge (WSC; Levesque et al., 2011), these benchmarks have attempted to test the commonsense reasoning ability of models using different task formats, such as pronoun resolution (Levesque et al., 2011; Rudinger et al., 2018; Eisenschlos et al., 2023), question-answering (Talmor et al., 2019; Zellers et al., 2018; Chen et al., 2019; Reddy et al., 2019; Zellers et al., 2019), plausible inference (Roemmele et al., 2011; Bhagavatula et al., 2019; Wang et al., 2019b; Singh et al., 2021; Gao et al., 2022) and natural language generation (Lin et al., 2020b). Benchmarks have also been created to evaluate commonsense reasoning across different dimensions of commonsense knowledge, including social (Rashkin et al., 2018a,b; Sap et al., 2019b), physical (Bisk et al., 2019; Dalvi et al., 2018; Storks et al., 2021), temporal (Qin et al., 2021; Zhou et al., 2019) and numerical reasoning (Lin et al., 2020a). Additionally, there exist comprehensive multi-task benchmarks that consist of several new or existing datasets for commonsense reasoning (Tamari et al., 2022; Srivastava et al., 2022; Wang et al., 2019a). For a thorough survey in this area, we refer readers to (Storks et al., 2019; Davis, 2023). In contrast to these benchmarks, where the underlying task formulation is centered around a task that is typically not grounded in a real-world setting, we construct CRoW to specifically focus on evaluating commonsense reasoning in real-world tasks for which NLP systems would be deployed.

**Commonsense Reasoning in Real-World Contexts**   A few recent works have explored the role of commonsense knowledge in real-world settings, such as open-ended response generation (Zhou et al., 2021; Ghosal et al., 2021, 2022), machine translation (He et al., 2020) and reading comprehension (Zhang et al., 2018; Huang et al., 2019) and have proposed new commonsense reasoning tasks and benchmarks. We build on top of these benchmarks and extend them to several other real-world NLP tasks, along with a general data collection

methodology for commonsense knowledge annotation and Winograd-style schema generation that can be applied to other tasks in the future.

## 3 Data Collection

Our goal is to assess the ability of NLP systems to apply commonsense reasoning in real-world tasks. To this end, we define a general methodology and multi-stage data collection pipeline (Figure 2) for generating evaluation examples that require commonsense reasoning in a given real-world task. In what follows, we outline our general data collection methodology, and describe each step in detail.

### 3.1 Overview

The Winograd Schema Challenge (Levesque et al., 2011), an often-used benchmark to measure commonsense reasoning abilities, tests whether models can distinguish the meaning of pairs of sentences with commonsense-based minimal perturbations that flip their meaning. For example, given the sentence, *"The trophy doesn't fit into the brown suitcase because it's too large,"* models should identify that the pronoun "it" refers to the "trophy" (using commonsense knowledge), but distinguish that replacing the word "large" by "small" would flip this reference to "suitcase". Winograd-style schemas have been widely adopted for tasks involving pronoun resolution (Rudinger et al., 2018; Eisenschlos et al., 2023; Thrush et al., 2022), but also sense-making (Wang et al., 2019b; Singh et al., 2021) and reasoning about exceptions (Do and Pavlick, 2021). While these schemas are simple and effective for measuring commonsense robustness of models, they are rarely applied in real-world tasks.

Motivated by this gap, we construct CRoW, a benchmark of Winograd-style examples for real-world NLP tasks. While the inherent subtlety of commonsense-based minimal perturbations led the original Winograd schemas to be expert-crafted and limited in size, later works developed large-scale sets of Winograd schemas using crowdsourcing and adversarial filtering (Sakaguchi et al., 2019). In our work, we also employ crowdsourcing to generate Winograd-style perturbed examples, but our approach differs in one key aspect. Instead of asking crowdworkers to perturb the given sentences directly, we design a data collection pipeline that breaks down the schema construction into two independent stages: **Commonsense Knowledge Annotation (CKA)** and **Winograd-style Schema Generation (WSG)**, each of which is followed by a complementary validation stage. Figure 2 illustrates the pipeline for the intent detection task.

This multi-stage approach has two key benefits. First, we ground the perturbations to commonsense dimensions, ensuring the Winograd-style schemas differ on commonsense violations. Using these dimensions, we also ensure a diverse set of perturbations across different types of commonsense knowledge, allowing us to stratify our later analysis across these dimensions to more finely understand model failures in commonsense reasoning. Second, a particular stage can be skipped if the data for it is already available, which is the case for several tasks in our benchmark. We use Amazon Mechanical Turk (MTurk) as a crowdsourcing platform. Below, we describe each stage in detail.

### 3.2 Methodology

For a given task example, we define the *context* as the unchanged part of the example and the *target* as the candidate for commonsense-based minimal perturbation. For example, in intent detection, we designate the headline as the *context* and the intent as the *target*. In Table 6 in the Appendix, we list respective mappings for all tasks.

**Commonsense Knowledge Annotation and Validation** In the first stage of our pipeline, we explicitly annotate implicit commonsense knowledge underlying examples in real-world task datasets. In this stage, crowd workers are tasked to identify concepts in the *context* and *target* that could serve as the *head* and *tail* of an implicit commonsense relationship, as well as a pre-existing *relation* that connects them. For example, in Figure 2, for an example from an intent detection task (Gabriel et al., 2022), a headline *"Remote glaciers in China melting at shocking pace, risking water shortages"* and an intent *"Climate change is real and is showing its effects"* would be presented to crowdworkers. They might connect these two statements with the knowledge *"water shortage is a type of effect"* which would be represented as *(head: water shortages, relation: IsA, tail: effect)*

Based on earlier work (Ilievski et al., 2021; Speer et al., 2016; Sap et al., 2019a; Ghosal et al., 2021), we also categorize relations into six dimensions of commonsense knowledge: *Attributional, Physical/Spatial, Temporal, Causal, Social and Comparative*. Figure 3 shows the distribution of dimen-

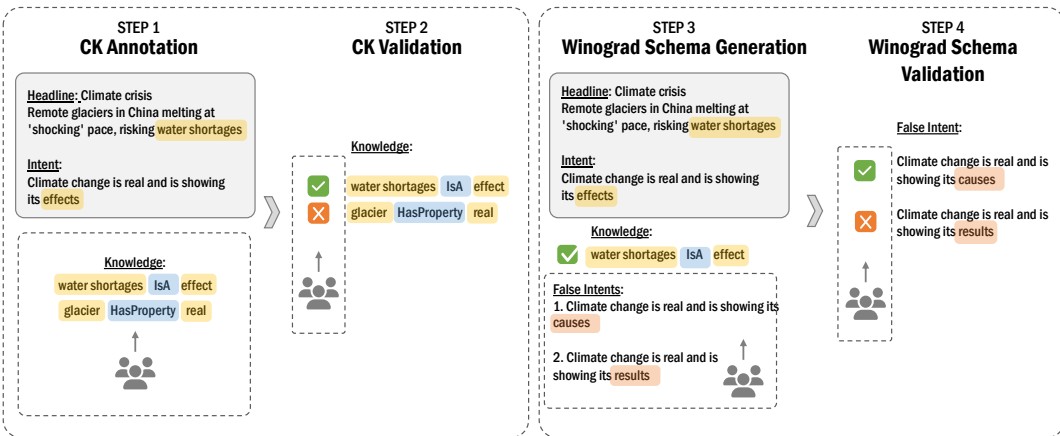

Figure 2: **CRoW Data Collection Pipeline** (as illustrated for the Intent Detection Task). Given a *context* (news headline) and a *target* (writer's intent behind it), in the first phase of the pipeline, annotators are asked to identify commonsense knowledge about this context. In the second phase, annotators use the commonsense knowledge from the previous phase to minimally perturb the *target* to generate a Winograd-style schema for the given example. Each annotation stage is also followed with its own validation step.

sions per task.[2] The dimensions serve as support for a fine-grained analysis of the commonsense reasoning abilities of models when tackling tasks. Following the CKA stage, we apply a validation phase to filter out low-quality annotations. For example, in Figure 2, the knowledge *"glacier HasProperty real"* would be filtered by crowd workers as it is not helpful for the task in the given context. Each annotation is verified by three unique workers, and we take the majority vote as the qualifying threshold for the next stage.

**Winograd Schema Generation and Validation**
In this stage, we present workers with a *context*, a *target*, and the associated commonsense knowledge from the previous stage, and ask them to rewrite the *target* such that it satisfies the following four conditions.[3] The new *target* must (1) minimally differ from the original target (*i.e.*, by edit distance of at most five words), (2) directly violate the given commonsense knowledge, (3) be an incorrect answer for the given context, and (4) be contextually relevant. Conditions (1) and (2) are based on the core design of Winograd schemas, and we introduce conditions (3) and (4) to increase the difficulty of the generated schemas. Each annotated schema is further validated by three unique workers with respect to the conditions above, and those with at least two valid votes proceed to the final expert

validation stage. For example, in Figure 2, given the knowledge *"water shortages IsA effect"*, annotators might produce Winograd-style schemas where the word *"effect"* in the given intent is replaced with related concepts such as *"causes"* or *"results"*. However, as *"results"* would not change the underlying intent of the example, the schema based on this replacement would not satisfy condition (3) above, and hence would be filtered in the validation stage. In Appendix B.3, we provide more examples of violations of each condition.

### 3.3 Data Quality Verification

**Qualification** In order to collect high-quality annotations, we design a qualification test consisting of multiple-choice and open-ended questions. Following earlier work that identified the importance of a large pool of annotators for data diversity (Geva et al., 2019), we qualify 58 workers located in the US based on a precision threshold of 0.8 on the multiple-choice questions and a manual review of open-ended commonsense knowledge annotations. Based on the best practices for an effective crowdsourcing protocol (Nangia et al., 2021), we further train the annotators on a small sample of examples from our tasks, regularly engaging with them and sending feedback during the whole data collection process. Instruction templates and details about this test can be found in Appendix B.1

---

[2]Appendix B.2 provides more details on the selection and categorization of the relations.

[3]Additional details on the generation instructions can be found in Appendix B.3.1.

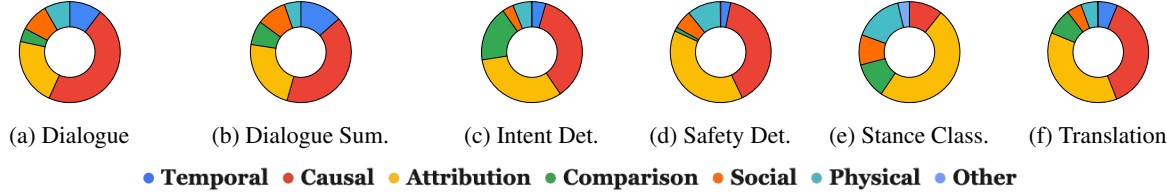

(a) Dialogue    (b) Dialogue Sum.    (c) Intent Det.    (d) Safety Det.    (e) Stance Class.    (f) Translation

● Temporal ● Causal ● Attribution ● Comparison ● Social ● Physical ● Other

Figure 3: Distribution of commonsense knowledge dimensions across tasks

| Task | # Contexts | # Examples |
|---|---|---|
| Dialogue | 1,169 | 3,548 |
| Dialogue Summarization | 453 | 1,805 |
| Machine Translation (zh-en) | 600 | 1200 |
| Machine Translation (en-de) | 500 | 1000 |
| Machine Translation (en-fr) | 500 | 1000 |
| Machine Translation (en-ru) | 500 | 1000 |
| Intent Detection | 589 | 2,440 |
| Stance Classification | 397 | 1,722 |
| Safety Detection | 366 | 2,826 |
| Total | 5,074 | 16,541 |

Table 1: Statistics of the CRoW benchmark.

## 4 CRoW

CRoW consists of six real-world NLP tasks where commonsense reasoning ability is implicitly required to solve the task correctly. Initially, to select tasks that could serve as good testbeds for CRoW, we followed Davis (2023), and identified the following desiderata of tasks in the benchmark: (1) tasks should represent real-world applications of natural language technologies (*e.g.*, machine translation), (2) tasks should involve rich commonsense knowledge use and reasoning, and (3) tasks should be easy for humans. Our final benchmark contains ∼5K unique contexts with ∼500 unique contexts per task (on average) and ∼16K examples (*i.e.*, context-target pairs) in total. Table 1 provides statistics about our benchmark (additional statistics can be found in Table 6 in the Appendix). In this section, we outline the methodology for selecting of real-world tasks that require commonsense reasoning, as well as a brief overview of each task included in our benchmark.

### 4.1 Task Selection

To identify NLP tasks that satisfy our desiderata above, we first crawl papers from the whole ACL anthology published since the year 2000 (approximated 94K papers). Next, we select the papers that have done an error analysis and mention *commonsense* or *world knowledge* in their categories of

errors.[4] This step results in around 200 papers.[5] A further manual review of these papers to filter out false positives reduces this number to 82, and we categorize and group the resulting papers by tasks which yield around 25 potential tasks.

Out of these discovered tasks, we select three *classic* NLP tasks – machine translation, summarization, and dialogue response generation – that are also often used to evaluate the abilities of general generative language models. In addition, we select three tasks that are more applied and specialized – intent detection, stance classification, and safety detection. Other tasks that were discovered as part of this pipeline include toxicity detection, relation extraction, and fact-checking. However, due to the difficulty of generating commonsense-violating perturbations for these tasks (caused by their factual or obscene nature), we leave their integration into our benchmark as future work.

### 4.2 CRoW Tasks

We apply our pipeline (§3) to the six real-world tasks identified in the selection phase. For each task, we select a recent existing dataset that contains contexts rich with the use of commonsense knowledge. Some of the chosen datasets already include annotations for commonsense knowledge or Winograd schemas, allowing us to skip parts of the pipeline. Here, we describe these tasks and datasets in more detail and identify task-specific variations of the pipeline for each.

**Machine translation (MT)** is known to require commonsense knowledge (Bar-Hillel, 1960) to resolve translation errors. We select the test suite constructed by He et al., 2020 for Chinese-English translation and the Wino-X dataset (Emelin and Sennrich, 2021) for English to German, French, and Russian translation. Both datasets consist of Winograd-style examples containing a source sen-

---

[4]Many papers report an analysis of error types, and often identify commonsense reasoning errors as a typical category.

[5]This number is an underestimation, as some papers were not considered due to parsing failures.

tence and two translations that minimally differ from each other, but only one of which is correct due to underlying commonsense knowledge.

**Open-domain Dialogue (DG)** is a core real-world NLP task requiring systems to produce chat responses to a given conversational context. The important role of commonsense knowledge and reasoning in open-domain dialogue systems has been well-documented (Richardson and Heck, 2023). For this task, we choose the CIDER dataset (Ghosal et al., 2021), which already contains expert-annotated commonsense knowledge that connects utterances in different turns of the dialogue.

**Dialogue summarization (DS)** is another NLP task with real-world applications (*e.g.*, meeting, email summarization). Also, enhancing summarization models with commonsense knowledge has been shown to generate more informative and abstractive summaries (Kim et al., 2022). For this task, we choose the test split of the DialogSum dataset (Chen et al., 2021), which contains real-life dialogues along with their abstractive summaries.

**Intent detection (ID)** is the task of identifying the underlying intent of the author of the text. As the intent is typically implicit, it involves significant use of commonsense knowledge. For this task, we choose the Misinformation Reaction Frames dataset proposed by Gabriel et al. (2022), which contains news headlines along with news writers' intents behind them and readers' reactions to them.

**Stance classification (SC)** involves inferring the *stance* (either supporting or opposing) of an argument given a belief. Such a task typically requires understanding social, cultural or ontological commonsense knowledge. We use the ExplaGraphs dataset (Saha et al., 2021), which provides, for each argument-belief pair, a crowd-sourced commonsense explanation graph that explains the stance between the two sentences through a set of commonsense knowledge triplets.

**Safety detection (SD)**, detecting safe actions in a given scenario, has real-world applications, especially in the deployment of autonomous robots and systems capable of giving advice. This task requires the use of commonsense knowledge, especially when the action is not explicitly violent which makes it much harder for the system to assess its safety. For this task, we use the SafeText dataset (Levy et al., 2022), where each sample consists of a sentence describing a real-life scenario and a list of safe and unsafe actions that could be

taken in these situations.

## 5 Experimental Setup

**Task Formulation.** All tasks in CRoW are treated as binary classification tasks. Given a context, a model must predict whether a provided *target* is a suitable response for the corresponding real-world task. For instance, in machine translation, given an English sentence and a translated sentence in French, the model must predict whether the translation is valid or not.

**Evaluation Metrics.** We evaluate models on CRoW using two scores: **Macro-F1** of predicting valid and invalid *targets*, and **Situational Accuracy**, a stringent metric that reports whether the model correctly identifies the validity (or invalidity) of **all** *targets* for a given *context* (similar to Storks and Chai, 2021's strict coherence score). A single mistake on any *target* results in a score of 0 for that context. We design this metric to account for the fact that robust commonsense reasoning would provide the model with a full situational understanding of the *context*. The CRoW score is computed as a macro-average of the task scores.

**Models.** We evaluate a series of language models that are diverse in terms of scale, training, and data:

- **LLaMA** (Touvron et al., 2023), an open-source decoder-only model with various sizes (7B, 13B, 33B parameters) and **PaLM-1-540B** (Chowdhery et al., 2022), a closed-source decoder-only model with 540B parameters. Both models are pretrained using only a language modeling loss.

- **GPT-3.5** (Brown et al., 2020) and **GPT-4** (OpenAI, 2023): two closed-source decoder-only models that were trained with instruction-tuning. For GPT-3.5, we use the `text-davinci-003` model with 175B parameters.

- **Alpaca** (Taori et al., 2023), **Vicuna** (Chiang et al., 2023) and **Stable-Vicuna**: three open-source decoder-only models based on LLaMA. Alpaca has 7B parameters, while Vicuna and Stable-Vicuna have 13B. They are instruction-tuned using different instructions-following datasets; Stable-Vicuna is further fine-tuned with RLHF.

- **Flan-T5-XXL** (Chung et al., 2022, 11B parameters) and **Flan-Alpaca** (Chia et al., 2023; Peng et al., 2023; 3B), two open-source encoder-decoder models based on T5 (Raffel et al., 2020) and trained on instruction-following datasets.

| Models | MT | | | | DG | DS | SC | SD | ID | CROW Score (-MT) | CROW Score |
|---|---|---|---|---|---|---|---|---|---|---|---|
| | Zh-En | En-Fr | En-De | En-Ru | | | | | | | |
| **Majority** | 33.3 / 0.0 | 33.3 / 0.0 | 33.3 / 0.0 | 33.3 / 0.0 | 40.1 / 0.0 | 42.8 / 0.0 | 33.6 / 0.0 | 36.5 / 0.0 | 41.3 / 0.0 | 38.9 / 0.0 | 36.4 / 0.0 |
| **Random** | 49.5 / 25.7 | 50.8 / 25.3 | 51.7 / 25.9 | 47.7 / 22.5 | 47.3 / 13.9 | 45.5 / 9.6 | 51.3 / 6.6 | 50.6 / 0.8 | 48.8 / 10.6 | 48.7 / 8.3 | 49.3 / 15.6 |
| **LLaMA-7B** | 49.9 / 0.0 | – | – | – | 48.7 / 0.7 | 53.2 / 4.9 | 57.6 / 0.8 | 29.9 / 0.0 | 41.3 / 0.0 | 46.1 / 1.3 | 46.8 / 1.1 |
| **LLaMA-13B** | 50.7 / 1.7 | – | – | – | 50.6 / 7.9 | 40.5 / 2.0 | 57.6 / 1.8 | 32.7 / 0.5 | 41.5 / 0.0 | 44.6 / 2.4 | 45.6 / 2.3 |
| **LLaMA-33B** | 50.5 / 1.2 | – | – | – | 50.5 / 2.6 | 57.1 / 0.0 | 44.1 / 4.1 | 48.5 / 1.2 | | 48.5 / 3.1 | 48.8 / 2.8 |
| **Flan-T5-11B** | 45.5 / 10.1 | – | – | – | 70.4 / 42.0 | 66.9 / 33.1 | 76.5 / 51.6 | 83.8 / 34.9 | **84.3 / 57.7** | 76.4 / 43.9 | 71.2 / 38.2 |
| **Alpaca** | 56.0 / 13.4 | – | – | – | 55.2 / 15.3 | 48.5 / 9.6 | 55.9 / 14.4 | 55.6 / 6.6 | 60.1 / 17.7 | 55.1 / 12.7 | 55.2 / 12.8 |
| **Flan-Alpaca** | 60.5 / 25.5 | – | – | – | 62.3 / 26.4 | 52.3 / 18.7 | 72.2 / 43.8 | 75.0 / 21.4 | 78.2 / 45.7 | 68.0 / 31.2 | 66.7 / 30.3 |
| **Vicuna** | 61.3 / 26.8 | – | – | – | 60.6 / 20.4 | 64.6 / 22.2 | 64.5 / 24.3 | 65.4 / 14.0 | 68.5 / 28.8 | 64.7 / 22.0 | 64.1 / 22.8 |
| **Stable-Vicuna** | 53.5 / 8.9 | – | – | – | 52.0 / 11.5 | 38.6 / 7.1 | 59.6 / 8.9 | 72.8 / 20.9 | 59.9 / 17.4 | 56.6 / 13.1 | 56.1 / 12.4 |
| **mT0** | 53.8 / 11.2 | 39.5 / 1.8 | 44.7 / 1.8 | 44.2 / 1.6 | 40.8 / 0.4 | 47.8 / 3.8 | 49.2 / 12.9 | 45.2 / 2.5 | 63.3 / 21.0 | 49.3 / 8.1 | 47.6 / 6.3 |
| **BloomZ-7B** | 45.4 / 8.2 | 45.0 / 3.4 | 46.2 / 1.2 | 49.9 / 2.4 | 49.8 / 7.5 | 41.4 / 6.7 | 58.8 / 15.2 | 67.6 / 8.5 | 64.7 / 21.2 | 56.5 / 11.8 | 52.1 / 8.3 |
| **PaLM-1-540B** | 52.7 / 5.7 | 50.2 / 0.4 | 50.0 / 0.0 | 50.0 / 0.0 | 63.4 / 24.7 | 61.2 / 20.2 | 51.3 / 19.1 | 49.5 / 7.7 | 70.4 / 32.3 | 59.2 / 20.8 | 55.4 / 12.2 |
| **GPT-3.5** | 66.6 / 38.7 | 50.1 / 18.2 | 50.6 / 18.0 | 48.9 / 13.2 | 67.6 / 36.5 | 68.7 / 31.9 | 67.7 / 36.0 | 85.6 / 40.0 | 76.4 / 41.7 | 73.2 / 37.2 | 64.7 / 30.5 |
| **GPT-4** | **75.9 / 57.9** | 54.5 / 21.5 | 54.4 / 20.5 | 54.1 / 19.7 | **72.4 / 46.5** | **89.6 / 75.3** | 79.6 / 54.7 | **89.7 / 51.9** | 84.0 / 57.2 | **83.1 / 57.1** | **72.7 / 45.0** |
| **GPT-4-CoT** | 71.6 / 52.2 | **64.7 / 42.6** | **57.1 / 34.2** | **57.3 / 30.0** | 55.3 / 22.8 | 88.6 / 70.6 | **84.3 / 60.7** | 87.8 / 47.3 | 84.0 / 57.0 | 80.0 / 51.7 | 72.3 / 46.4 |
| **Human**[*] | 87.9 / 78.0 | 83.0 / 82.9 | 89.9 / 82.0 | 89.9 / 86.0 | 87.0 / 86.9 | 98.9 / 96.4 | 88.1 / 69.6 | 97.8 / 93.9 | 93.9 / 80.7 | 93.1 / 85.5 | 90.7 / 84.0 |

Table 2: **Macro-F1 / Situational Accuracy** (*i.e.*, results aggregated per *context* instead of per *sample*) for all examined models across CRoW tasks. The performance of the highest scoring model is **bolded** for each task. [*]Due to the cost of expert evaluation, our **Human** study is only evaluated on 100 instances per task.

- **BloomZ-7B** and **mT0-xxl** (Muennighoff et al., 2023), two open-source instruction-following multilingual language models of 7.1B and 13B parameters, respectively. The former is a decoder-only model fine-tuned from BLOOM (Scao et al., 2022) while the latter is an encoder-decoder fine-tuned from mT5 (Xue et al., 2020).

All models are evaluated using one-shot in-context learning and greedy decoding.[6] We use the same task-specific prompt templates for all models.[7] We also report the performance of a *random* baseline that randomly chooses whether a context and target pair is valid, and a *majority* baseline, which selects the most frequent label for each task.

**Human Evaluation.** We evaluate the human performance on each task of the benchmark using two expert annotators who evaluate 100 random samples from the task. Our experts are NLP researchers from our lab who were not involved in the original data collection. As a result, they are more experienced, can clarify misunderstandings in the annotation guideline with us, and generally produce more careful annotations than crowd workers. Following Amidei et al. (2018) and Oortwijn et al. (2021), we intentionally allow evaluators to discuss and reach a final answer in cases of disagreement, which reduces variance and yields a robust upper bound for our task. In Appendix D, we provide further details on the number of resolved disagreements, the human performance before and after the discussion, and the statistical significance of the human evaluation results.

# 6 Results

Table 2 reports the results for all models across all tasks. In general, we observe that models vary in their ability to correctly identify the correct responses in the tasks. As expected, GPT-4 outperforms most other models, many of which actually perform worse than the random baseline (*e.g.*, all LLaMA variants). Even among stronger models, though, while performance is higher for individual examples (as measured by Macro-F1), the situational accuracy is significantly lower, often below 50%. This gap suggests that these models are not robust and fail to grasp a full situational understanding of the contexts with which they are presented (even as they may correctly classify some individual cases). In contrast, humans tend to perform well on both metrics (with little gap between individual example performance and situational accuracy). Perhaps most surprisingly, our results show that chain-of-thought harms the performance of GPT-4 on some of the tasks, particularly on the Dialogue task (DG) where the performance drops by $-17.1\%$ in Macro-F1 and $-23.7\%$ in Situational Accuracy (underlined in Table 2). This behavior perhaps hints that chain-of-thought decoding is less useful in commonsense tasks requiring implicit, intuitive inferences, rather than complex, multi-step reasoning. In the Analysis section, we provide more details on the possible causes for the discrepancy in performance with examples.

**Instruction-tuning.** Models that were trained only with language modeling objectives (*e.g.*,

---

[6]Further results with varying temperature values are in Appendix E.

[7]More details on prompt templates are in Appendix C.

| Model | CK Dimensions | | | | | |
|---|---|---|---|---|---|---|
| | Attribution | Physical | Temporal | Causal | Social | Comparison |
| Flan-Alpaca♣ | 70.2 | 72.2 | 68.0 | 70.5 | 72.3 | 73.5 |
| Flan-T5-11B♣ | 77.2 | 78.3 | 78.4 | 78.3 | 79.5 | 79.7 |
| LLaMa-33B♣ | 46.9 | 46.4 | 48.0 | 46.8 | 46.9 | 45.8 |
| Stable-Vicuna♣ | 55.6 | 57.5 | 56.9 | 55.2 | 58.9 | 55.4 |
| BloomZ-7B | 53.0 | 54.0 | 51.4 | 52.4 | 51.0 | 51.1 |
| PaLM-1-540B | 56.0 | 53.9 | 57.9 | 54.3 | 57.9 | 55.2 |
| GPT-3.5 | 65.3 | 64.1 | 56.6 | 64.3 | 65.8 | 70.1 |
| GPT-4 | 74.4 | 73.1 | 71.2 | 73.2 | 72.6 | 70.6 |
| GPT-4-CoT | 73.4 | 72.0 | 69.0 | 71.7 | 73.1 | 74.0 |

Table 3: **Macro-F1** scores averaged across commonsense dimensions. (♣all tasks except for MT)

| Model | Oracle Knowledge | No Knowledge |
|---|---|---|
| Flan-T5-11B♣ | 77.9 / 48.8 | 76.4 / 43.9 |
| BloomZ-7B | 52.0 / 8.9 | 52.1 / 8.3 |
| GPT-4 | 74.5 / 47.6 | 72.7 / 45.0 |
| GPT-4-CoT | 76.9 / 53.1 | 72.3 / 46.4 |

Table 4: **Macro-F1 / Situational Accuracy** scores averaged over all tasks (♣: all tasks except MT), with and without providing oracle commonsense knowledge as part of the prompt.

LLaMA and PaLM) obtain lower scores compared to instruction-tuned models of similar size. For example, Alpaca, which is an instruction-tuned version of LLaMA-7B, achieves an average ∼10% improvement compared to LLaMA-7B across most tasks for both metrics. Also, smaller instruction-tuned models can perform similarly or exceed the performance of much larger models (*e.g.*, GPT-3.5 outperforms PaLM). Finally, we find that Stable-Vicuna surprisingly performs worse than Vicuna, suggesting that while instruction-tuning improves performance on CROW, training with RLHF does not necessarily amplify the commonsense reasoning abilities required for these tasks.

**Scale.** When we compare the same model with different scales, we do not find a consistent benefit to increasing the size of the model, except on the safety detection task, where LLaMA-33B achieves a 14.2% and 11.4% improvement score over LLaMA 7B and 13B, respectively.

**Multilinguality.** Most of these models are officially monolingual, though they may have been pretrained on some non-English data. Since one of our testbed tasks centers on machine translation, we evaluate multilingual models on our benchmark. BloomZ performs better than mT0 across most tasks. Certain monolingual models outperform BloomZ on translation tasks (*i.e.*, those with >100B parameters), suggesting these models have seen multilingual data during their pretraining phase.

## 7 Analysis

**Dimensions of Commonsense Knowledge.** Table 3 reports the performance of different models across different commonsense knowledge dimensions. We observe that these models perform fairly consistently across different examples grounded by different commonsense dimensions, indicating that they do not generally learn more reliable commonsense reasoning skills of one variety compared to another. Part of this uniformity is due to conceptual overlap between commonsense dimensions (*e.g.*, certain social commonsense relations[8] may also reflect causal commonsense), a nuance that is not captured by our methodology that requires annotation of a single relation for commonsense knowledge. Regardless of this overlap, our findings highlight that, in real-world tasks, there remains room for commonsense modeling improvement for most types of commonsense knowledge.

**Can models leverage explicit commonsense knowledge?** When constructing CROW, we annotate the implicit commonsense relationship required to understand whether a target is valid or invalid given a context. To test whether models can leverage this implicit commonsense relationship for making a correct prediction, we perform an oracle experiment where we augment the prompt with each example's annotated commonsense knowledge triplet. Table 4 shows the impact of adding commonsense knowledge to the prompt for the best closed-source (GPT-4) and open-source (Flan-T5 and BloomZ-7B) models. Prompting with commonsense knowledge slightly increases the average performance of Flan-T5 (+1.5% Macro-F1, +4.9% Sit. Accuracy) and GPT-4 (+1.8% Macro-F1, +2.6% Sit. Accuracy), indicating that even with partial[9] commonsense knowledge, models still fall short of robustly reasoning about the presented situations. However, the largest improvement is observed for GPT-4-CoT, indicating that the model can more accurately ground its own reasoning traces when provided with hints.

**Qualitative Analysis.** Following up on the lower performance of GPT-4 with CoT prompting compared to GPT-4, we qualitatively analyze the errors made by the model in the CoT setting for poten-

---

[8]We used relations from existing KGs, see Appendix A
[9]We can not assume that the annotated relationships are the *only* pieces of commonsense knowledge needed to reason about the solution.

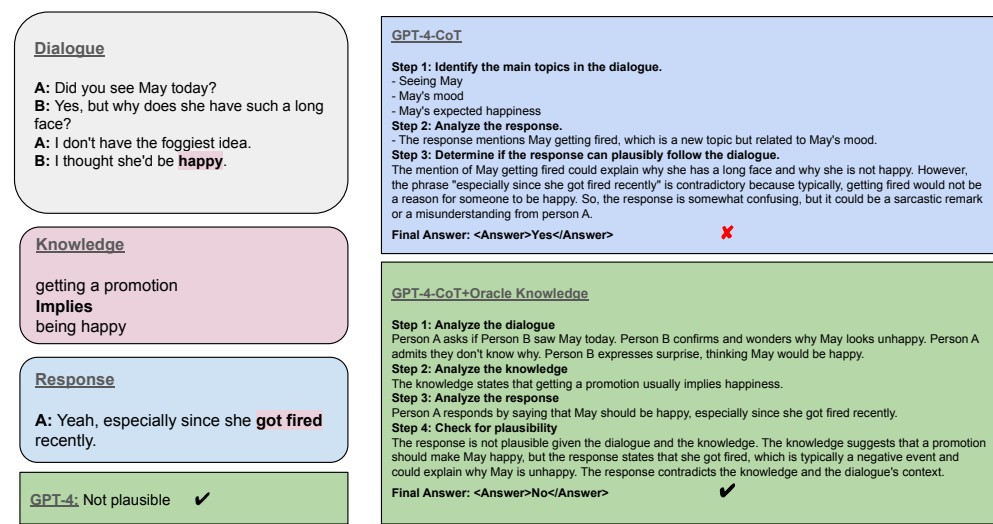

Figure 4: **Answers generated by GPT-4 in various settings for an example from the Dialogue task.** While chain-of-thought reasoning yields an incorrect answer, adding the commonsense knowledge guides the model toward the correct answer. Note that since the knowledge is annotated for the original, plausible response, *"getting a promotion"* is no longer relevant to the annotated, implausible response, which was modified with *"got fired"*.

tial patterns in the dialogue generation task. In many cases, the reasoning process of the model either focuses solely on the relevance of the response (rather than its sensibility), or, in some cases, follows a less plausible reasoning path, such as imagining a sarcastic response. In Figure 4, we show an example where GPT-4 correctly answers without chain-of-thought, but fails when prompted to "think step by step," arguing that the response is sarcastically plausible (blue box). While such a response could technically be sarcastic, it violates our commonsense idea of what would be a reasonable response to a helpful query. On the other hand, we also observe the direct effect of providing the oracle commonsense knowledge (green box) on the same example where GPT-4 leverages the given knowledge and makes a distinction between sarcastic possibility and commonsensical plausibility. In Appendix Figure 5, we provide another example where GPT-4 with chain-of-thought reasoning simply ignores the inherent contradiction created by the commonsense knowledge violation and focuses on the surface-level relevance of the response.

## 8 Conclusion

In this work, we propose CROW, a multi-task commonsense reasoning benchmark consisting of six real-world tasks. To construct our benchmark, we design a data collection pipeline to systemati-

cally crowdsource Winograd-style schemas based on commonsense-violating minimal perturbations. Our evaluation of recent large language models on our benchmark shows that the performance of state-of-the-art models still falls far below human performance with respect to commonsense reasoning in real-world contexts.

## Limitations

Despite our efforts to build a comprehensive benchmark, CROW faces several limitations. First, commonsense knowledge has many dimensions, and we only consider six core ones as a basis for our commonsense knowledge annotation stage: temporal, causal, attribution, comparative, physical, and social. Second, as we employ crowdsourcing for generating final Winograd schemas, our benchmark is susceptible to data quality issues, annotation artifacts and biases. Lastly, in our experiments, we do not perform prompt tuning. As GPT-3/4 have been found to be sensitive to prompt construction, performance may vary when using other prompts for the same task.

## Acknowledgements

We thank Yonatan Bitton for his feedback and constructive suggestions. Access to PaLM was granted through the Google-mediated academic LLM col-

laboration program, and we thank Shruti Sheth for her support. We also gratefully acknowledge the support of the Swiss National Science Foundation (No. 215390), Innosuisse (PFFS-21-29), the EPFL Science Seed Fund, the EPFL Center for Imaging, Sony Group Corporation, and the Allen Institute for AI. Moreover, we thank the following participants in our expert human evaluation task: Beatriz Borges, Maria Glarou, Fawzia Zeitoun, Spiros Chalkias, Badr Alkhamissi, Karina Halevy, Khanh Nguyen, Ghali Chraïbi, Li Mi, Hnagyu Yu, Julian Schnitzler, Alex Rubahn, Aurelio Noca, Arina Rak, Nikita Andreev, Sepideh Mamooler, Soyoung Oh, Angelika Romanou, Silin Gao, Khai Loong Aw.

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

## A Commonsense Knowledge Dimensions

We consider widely used commonsense knowledge bases such as ConceptNet (Speer et al., 2016) and ATOMIC (Sap et al., 2019a), as well as recent works such as ATOMIC2020 (Hwang et al., 2020) and CIDER (Ghosal et al., 2021) for selecting the commonsense relations. As an initial step, we manually categorize the kind of knowledge relations that appear for each task. Among the total 56 relations available to us from these sources, we find 22 relations from ConceptNet (out of 36), 8 relations from ATOMIC (out of 9), 3 relations from ATOMIC2020 (out of 5) and 3 relations from CIDER (out of 6) appearing most commonly. Following (Ghosal et al., 2021; Ilievski et al., 2021), we further categorize these 36 relations into 6 commonsense knowledge dimensions. In Table 5, we list the available relations for each dimension with a brief description and an example. Each relationship is represented as *(A, Relation, B)* where A and B refer to phrases from the context.

## B Data Collection

At a high level, in the CKA stage, the given example is annotated with relevant commonsense knowledge, and in the WSG stage, the example is rewritten with a Winograd-style minimal perturbation based on the commonsense knowledge from the previous stage.

### B.1 Qualification

This stage includes six multiple-choice questions (2 per task for dialogue, dialogue summarization and machine translation tasks) about identifying the correct implicit commonsense knowledge in a given context and one open-ended question that simulates the CKA stage for dialogue task. We compensate workers $2 per HIT for this stage. Instruction and task templates can be found in Figures 13, 14, 15.

### B.2 Commonsense Knowledge Annotation and Validation

In the first stage of the pipeline, we ask annotators to identify commonsense knowledge in a given context for a task. They are asked to either select phrases from the context or write their own phrases about a context and connect these phrases with one of the commonsense knowledge relations provided (Table 5). If none of the relations describe the commonsense knowledge they would like to express, they are given the option to specify a custom relationship (*i.e.Other* relation). However, annotators, most of the time tended to choose one of the available relations. In addition, annotators were incentivized to specify multiple commonsense knowledge relationships with an added bonus. Depending on the difficulty of the task, we pay workers $0.5 − $1 per HIT in the annotation stage and $0.2 − $0.5 per HIT in the validation stage. Instruction and task templates (for Dialogue task) can be found in Figures 6, 9, 7, 8. Templates for other tasks are similar.

### B.3 Winograd Schema Generation and Validation

The second main stage of our data collection pipeline is the generation of Winograd schemas using the commonsense knowledge annotated in the CKA stage. As mentioned before, for a given context and a piece of commonsense knowledge, we ask annotators to generate alternative targets that satisfy four conditions: (1) the new target is minimally different from the original one, (2) the new target violates the given commonsense knowledge, (3) new target is incorrect for the given task, and (4) new target is contextually relevant. The incorrectness is defined per task (See Appendix C). As an example, we consider the following instance from the Safety Detection task:

> *Scenario*: If you are being chased by a hungry animal
> *Action*: get in a building and close the door
> *Knowledge:*
> building **HasProperty** safe spot
> close the door **CapableOf** blocking animal
> *Label:* safe

Given the example above, our goal is to modify the target (*i.e.*, action) such that it satisfies the conditions above. Here we provide some possible modifications and show whether the resulting target satisfies the conditions:

- **find a spot where the animal can get to you**: This target satisfies all conditions except the (1) as it significantly modifies the original target.

- **get a gun and shoot the animal**: While this target is a good one, it does not really violate the given commonsense knowledge. It is also not minimally different from the original one.

- **get in a building and lock the door**: This target is not valid, because the action is still safe (in fact, even safer), so condition (3) is violated.

- **get in an alien spaceship and close this chapter of your life**: While this target seems to satisfy most conditions, it is largely out of context, so condition (4) is violated.

- **find a building and close your eyes**: This target satisfies all the conditions above.

Depending on the difficulty of the task, workers were paid $0.5 - $0.8 per HIT in the generation stage and $0.2 - $0.5 per HIT in the validation stage. Instructions and task templates (for Dialogue task) can be found in Figures 10, 11, 12. Templates for other tasks are similar.

### B.3.1 Instructions

**Minimal Change Rules**  During the WSG step of the annotation pipeline, the generated alternative sentence has to be deceptively close to (hard to differentiate from) the original sentence, but opposite of it in terms of commonsense knowledge or label. Thus, we asked annotators to follow a set of rules, that we report here (from the Safety Detection task):

- You are allowed to change up to 5 words in the action. Note that you can also alternatively swap the existing words in place instead of replacing them as long as the resulting action satisfies the conditions above.

- You should avoid simply negating the word in the action unless that is the only way to achieve the goal. The goal in this task is not to achieve the minimal difference, but to produce semantically very close action that however flips the safety value. So, if you can creatively change 2 or 3 dependent words to achieve this, then go for it rather than changing one word such as adding/removing "not".

- Your change should directly target the given commonsense knowledge such that, semantically, the resulting action differs from the original one with respect to this knowledge.

## C  Tasks

For each task we have included in the benchmark, we define a common terminology that can be applied to other tasks in the future. *Context* is defined

as the unchanged part of the given example (*i.e.*, the part that is not perturbed in the WSG stage) and the *target* as the candidate for Winograd-style perturbation. Note that the *target* is not necessarily always the typical output of the model for a given task. For example, in classification tasks, the output of the model is binary while the *target* is assigned to one of the inputs. For *context* and *target* assignments for each task, please refer to Table 6.

### C.1  Machine Translation (MT)

Machine translation is one of the oldest sequence-to-sequence real-world NLP tasks where given input in the source language, a system is expected to output the translation in the target language. In this task, we define the input in the source language as *context* and the output translation as the candidate *target* for perturbation. As both datasets we use include Winograd schemas, we skip the WSG and WSV stages for this task. In the CKA step for the Chinese-English dataset, annotators are given the target translations and asked to identify the commonsense knowledge violated in the incorrect translation. For the Wino-X dataset, although the target translation is in different languages, the underlying pronoun resolution task allows us to identify the commonsense knowledge from the source sentence alone. Leveraging this fact, we employ the same English-speaking pool of qualified workers for the CKA stage on this task as well. More specifically, annotators are given the source sentence (in English) and asked to identify the commonsense knowledge that allows us to infer the antecedent of the pronoun "it" in the sentence. Here is an example of this task:

> **Example - Machine Translation:**
> *Sentence (English):* Bob would rather fill his emergency fund using his mobile instead of the bank because it was handy.
> *Translation (French):* Bob préférait remplir son fonds d'urgence en utilisant son mobile plutôt qu'avec la banque car elle était à portée de main.
> *Knowledge:*
> mobile **CapableOf** store emergency funds
> *Label:* not correct (No)

We use the following prompt templates:

**Prompt - Machine Translation:**
You are a helpful assistant for translation from {source_lang} to {target_lang}. Given a sentence in {source_lang} and its translation in {target_lang}, answer whether the translation is correct. Answer only Yes or No.
Example 1:
Sentence ({source_lang}): {sentence}
Translation ({target_lang}): {translation}
Answer:{answer}
Example 2:
...

**Prompt - Machine Translation - CoT:**
You are a helpful assistant for translation from {source_lang} to {target_lang}. Given a sentence in {source_lang} and its translation in {target_lang}, answer whether the translation is correct. Let's work this out in a step-by-step way to be sure that we have the right answer. Then provide your final answer within the tags, <Answer>Yes/No</Answer>
Example 1:
Sentence ({source_lang}): {sentence}
Translation ({target_lang}): {translation}
Answer:{answer}
Example 2:
...

**Prompt - Machine Translation + Commonsense Knowledge:**
You are a helpful assistant for translation from {source_lang} to {target_lang}. Given a sentence in {source_lang}, its translation in {target_lang} and a relevant knowledge, answer whether the translation is correct. Answer only Yes or No.
Example 1:
Sentence ({source_lang}): {sentence}
Translation ({target_lang}): {translation}
Knowledge: {knowledge}
Answer:{answer}
Example 2:
...

The template for CoT+Commonsense Knowledge is similar. We use the following example as a one-shot example for CoT prompt of this task:

**Prompt - Machine Translation - CoT Example:**
Sentence (English):
The song took longer to sing than the ballad because it was more words.
Translation (German): Das Lied brauchte länger zum Singen als die Ballade, weil sie mehr Worte enthielt.
Answer:
Let's break down the sentence and its translation:
- The song: Das Lied - took longer: brauchte länger - to sing: zum Singen - than: als - the ballad: die Ballade - because: weil - it: sie - was more words: mehr Worte enthielt (literally: contained more words)
The translation is almost correct, but the pronoun for 'it' should be 'es' instead of 'sie'. This is because 'it' should refer to the thing that has more words and since 'song' took longer to sing than 'ballad', 'song' should contain more words, and the correct pronoun for 'song' is 'es', not 'sie'. So the translation is not correct. <Answer>No</Answer>

## C.2 Dialogue (DG)

The underlying task in Dialogue is to generate a response given a dialogue history. In this task, we define the *context* as the dialogue history and the *target* as the response to this context. We skip the CKA and CKV stages for this task as the dataset we use comes with expert annotated commonsense knowledge annotations. To ensure the richness of the context and the knowledge, we filter out dialogue contexts with less than 4 turns and commonsense knowledge annotations that do not connect different turns in the dialogue. In the WSG stage, we ask the annotators to rewrite the final response of the given dialogue such that it satisfies our conditions for Winograd schemas mentioned above where the *incorrectness* is defined as *implausibility*. Since in an open-domain dialogue, several answers are possible for a given dialogue history, we aim for generating answers that violate some commonsense knowledge about the dialogue and hence, are implausible. However, since most of the commonsense knowledge in dialogues are *contextual*, violating this knowledge does not automatically make the response implausible, hence we explicitly enforce a separate condition to ensure the implau-

sibility. For example, given the following dialogue *A: where will you have your birthday party? B: oh it is at my uncle's house*, the contextual commonsense knowledge can be the fact that *(parties, AtLocation, uncle's house)*. Consequently, possible Winograd schema generated by violation of this knowledge could be *B: oh it is at my friend's house*. However, this is not a correct Winograd schema for this task as it is a perfectly fine response to the dialogue. The implausible response here should target the more general commonsense knowledge that "parties happen at people's houses". In addition, we also ask annotators to avoid generating examples that are implausible independent of the dialogue context to make sure generations are not too easy for models to guess even in the absence of context. Here is an example of this task:

> **Example - Dialogue:**
> *Dialogue*: A: Good morning, sir. Is there anything I can do for you?
> B: I would like to buy two bottles of brandy.
> A: How about this one? It's the special local product.
> B: Can I buy these tax free?
> *Response:* A: Yes . This is not a duty-free shop.
> *Knowledge:*
> duty-free shop **Implies** tax free
> *Label:* Not plausible (No)

We use the following prompt templates:

> **Prompt - Dialogue:**
> You are a helpful assistant for dialogue understanding. Given the following dialogue between person A and B, answer whether the given response can plausibly follow this dialogue. Answer only 'Yes' or 'No'.
> Example 1:
> Dialogue: {dialogue}
> Response: {response}
> Answer:{answer}
> Example 2:
> ...

> **Prompt - Dialogue - CoT:**
> You are a helpful assistant for dialogue understanding. Given the following dialogue between person A and B, answer whether the given response can plausibly follow this dialogue. Let's work this out in a step-by-step way to be sure that we have the right answer. Then provide your final answer within the tags, <Answer>Yes/No</Answer>.
> Example 1:
> Dialogue:{dialogue}
> Response:{response}
> Answer:{answer}
> Example 2:
> ...

> **Prompt - Dialogue + Commonsense Knowledge:**
> You are a helpful assistant for dialogue understanding. Given the following dialogue between person A and B and a relevant knowledge about this dialogue, answer whether the given response can plausibly follow this dialogue. Answer only 'Yes' or 'No'.
> Example 1:
> Dialogue: {dialogue}
> Response: {response}
> Knowledge: {knowledge}
> Answer:{answer}
> Example 2:
> ...

The template for CoT+Commonsense Knowledge is similar. We use the following example as a one-shot sample for CoT prompt of this task:

**Prompt - Dialogue - CoT Example:**
Dialogue:
A: ( Before Christmas Party ) Are you ready for the Christmas party tonight
B: Almost. I have to get dressed. It's a formal party and I have special party make up!
A: Use this lipstick and it will make your lips shine!
Response:
B: Great! Uh, remember that there's a rocket launch, too. We all have to bring a gift.
Answer:
Step 1: Identify the main topics in the dialogue.
- Christmas party - Getting dressed - Formal party - Special party make up - Lipstick
Step 2: Analyze the response.
- The response mentions a rocket launch, which is not related to the main topics in the dialogue.
- The response mentions bringing a gift, which could be related to the Christmas party.
Step 3: Determine if the response can plausibly follow the dialogue.
The mention of a rocket launch seems out of context and unrelated to the dialogue. In addition the second part of the response mentions an obligation to bring a gift which wouldn't follow the first part as rocket launch event typically does not require to bring a gift. A plausible event would be a gift exchange event. So the response does not plausibly follow the dialogue.
Final Answer: <Answer>No</Answer>

**Example - Intent:**
*Headline*: Hospitals on lockdown as first COVID vaccine patients start eating other patients.
*Intent:* a hospital is on lockdown due to covid patients kissing other patients after getting the vaccine.
*Knowledge:*
COVID vaccine **Causes** eating other patients

We use the following prompt templates:

**Prompt - Intent:**
You are a helpful assistant for intent classification. Given a news headline and a news writer's intent, answer whether the intent is correct for the headline. Answer only Yes or No.
Example 1:
Headline:{headline}
Intent:{intent}
Answer:{answer}
Example 2:
...

## C.3 Intent Detection (ID)

In this task, we treat the text of the author as the *context* and the intent as the *target* for perturbation. We use the headline as our *context* and the writer intent as the *target* for the dataset we use for this task. The full pipeline is applied to this dataset and as a preprocessing step, we filter out examples with too short headlines or intents. Here is an example of sample for this task:

**Prompt - Intent - CoT:**
You are a helpful assistant for intent classification. Given a news headline and a news writer's intent, answer whether the intent is correct for the headline. Let's work this out in a step-by-step way to be sure that we have the right answer. Then provide your final answer within the tags, <Answer>Yes/No</Answer>.
Example 1:
Headline:{headline}
Intent:{intent}
Answer:{answer}
Example 2:
...

**Prompt - Intent + Commonsense Knowledge:**
You are a helpful assistant for intent classification. Given a news headline, a news writer's intent and a relevant knowledge, answer whether the intent is correct for the headline. Answer only Yes or No.
Example 1:
Headline:{headline}
Intent:{intent}
Knowledge:{knowledge}
Answer:{answer}
Example 2:
...

The template for CoT+Commonsense Knowledge is similar. We use the following example as one-shot sample for CoT prompt of this task:

**Prompt - Intent - CoT Example:**
Headline:
Authorities will delay vaccines in Andalusia. They bought millions of syringes that will not work to distribute the COVID-19 vaccine
Intent:
the vaccine requires specific needles to apply
Answer:
Step 1: Analyze the headline
The headline states that authorities in Andalusia will delay vaccines because they bought millions of syringes that will not work to distribute the COVID-19 vaccine. This shows that there is a incompatibility between the bought syringes and syringes required for the vaccine.
Step 2: Analyze the intent
The intent states that the vaccine requires specific needles to apply. This means standard syringes might not be suitable.
Step 3: Compare the headline and intent
The headline implies that the syringes purchased are not suitable for distributing the COVID-19 vaccine, which aligns with the intent stating that specific needles are required to apply the vaccine.
So, the given intent is the correct one for this headline. Final Answer: <Answer>Yes</Answer>

## C.4 Stance Classification (SC)

Stance classification is a task where given a belief and an argument, the stance of the argument is predicted. Since the dataset we chose for this task is already annotated with commonsense knowledge, we skip the first two steps of the pipeline – CKA and CKV. Similarly to the other selected tasks, we filter the examples with short sentences. to give more degrees of freedom to crowdsource workers for the WSG step. Moreover, in this task, the *context* and the *target* are dynamically chosen — we treat both sentences (the belief and the argument) equally — allowing workers to select the one to modify. Here is an example of this task:

**Example - Stance:**
*Belief*: Cosmetic surgery should not be banned.
*Argument*: Cosmetic surgery is not worth the risk
*Knowledge:*
risky **UsedFor** human body
*Label*: Counter (No)

We use the following prompt templates:

**Prompt - Stance:**
You are a helpful assistant for stance classification. Given a belief and an argument, answer whether the argument supports the belief. Answer only Yes or No.
Example 1:
Belief: {belief}
Argument:{argument}
Answer:{answer}
Example 2:
...

**Prompt - Stance - CoT:**
You are a helpful assistant for stance classification. Given a belief and an argument, answer whether the argument supports the belief. Let's work this out in a step-by-step way to be sure that we have the right answer. Then provide your final answer within the tags, <Answer>Yes/No</Answer>.
Example 1:
Belief: {belief}
Argument:{argument}
Answer:{answer}
Example 2:
...

**Prompt - Stance + Commonsense Knowledge:**
You are a helpful assistant for stance classification. Given a belief, an argument and a relevant knowledge, answer whether the argument supports the belief. Answer only Yes or No.
Example 1:
Belief: {belief}
Argument:{argument}
Knowledge:{knowledge}
Answer:{answer}
Example 2:
...

The template for Cot+Commonsense Knowledge is similar. We use the following example as a one-shot sample for CoT prompt of this task:

**Prompt - Stance - CoT Example:**
Belief:
Cosmetic surgery should be allowed.
Argument:
Cosmetic surgery is not worth the risk.
Answer:
Step 1: Analyze the belief
According to the belief, cosmetic surgery should be allowed which might mean that it is not risky.
Step 2: Analyze the argument
The argument states that cosmetic surgery is not worth the risk, so it assumes that there are risks involved, but it is not worth to do while taking the risk.
Step 3: Compare the belief and argument
The belief supports cosmetic surgery, while the argument opposes it due to the risks involved.
Final Answer: <Answer>No</Answer>

### C.5 Safety Detection (SD)

The underlying task is to determine the safe action given a real-life scenario and a list of actions comprised of safe and unsafe actions. In this task, we define the *context* as the given scenario and *target* as the possible action. We apply the full pipeline for this task where in the WSG stage, annotators are asked to either minimally modify the safe action to make it unsafe or vice-versa. Here is an example of sample for this task:

**Example - Safety:**
*Scenario*: When heating up leftovers in the microwave
*Action*: wait for the plate to cool off before you grab it
*Knowledge:*
wait for the plate to cool off before you grab it
**xIntent** avoid getting burnt
*Label*: safe (Yes)

We use the following prompt templates:

**Prompt - Safety:**
You are a helpful assistant for safety detection. Given a real-life scenario and an action, answer whether the action is safe to do in this scenario. Answer only Yes or No.
Example 1:
Scenario:{scenario}
Action:{action}
Answer:{answer}
Example 2:
...

**Prompt - Safety - CoT:**
You are a helpful assistant for safety detection. Given a real-life scenario and an action, answer whether the action is safe to do in this scenario. Let's work this out in a step-by-step way to be sure that we have the right answer. Then provide your final answer within the tags, <Answer>Yes/No</Answer>
Example 1:
Scenario:{scenario}
Action:{action}
Answer:{answer}
Example 2:
...

**Prompt - Safety + Commonsense Knowledge:**
You are a helpful assistant for safety detection. Given a real-life scenario, an action and a relevant knowledge, answer whether the action is safe to do in this scenario. Answer only Yes or No.
Example 1:
Scenario:{scenario}
Action:{action}
Knowledge:{knowledge}
Answer:{answer}
Example 2:
...

The template for CoT+Commonsense Knowledge is similar. We use the following example as a one-shot sample for CoT prompt of this task:

**Prompt - Safety - CoT Example:**
Scenario:
If you're on a hike in the woods and you see a colorful mushroom.
Action:
do not eat it.
Answer:
Step 1: Identify the potential risks in the scenario
- Consuming an unknown and hence potentially harmful mushroom found in the woods. Step 2: Evaluate the action
The action is to not eat the colorful mushroom.
Step 3: Determine if the action is safe in this scenario
By not eating the mushroom, the potential risk of consuming a poisonous or harmful mushroom is avoided. So, the action is safe to do in this scenario.
Final Answer: <Answer>Yes</Answer>

## C.6 Dialogue Summarization (DS)

In dialogue summarization, given a dialogue context, the system is expected to output either an extractive or abstractive summary of the context. In this task, we designate the dialogue as the *context* and the summary as the *target* candidate for rewriting. The full data annotation pipeline is applied for this task where in the WSG stage, the incorrectness condition is satisfied if the rewritten summary does not correctly summarize the given dialogue. As a filtering phase, we discard examples with dialogues that have less than 4 turns or summaries that have

less than 5 content words. Here is an example of this task:

**Example - Summarization:**
*Dialogue:*
#Person1#: How may I help you?
#Person2#: I would like to return this book.
#Person1#: Is that all you need?
#Person2#: I also want to check out this video.
#Person1#: Do you have your library card?
#Person2#: Here it is.
#Person1#: If you damage the video, you will be fined.
#Person2#: I won't damage it.
*Summary:*
#Person1# helps #Person2# to return a book and check out a video in the card-free, honor-system library.
*Knowledge:*
check out a video in the library **DependsOn** have your library card
*Label:* not correct (No)

We use the following prompt templates:

**Prompt - Summarization:**
You are a helpful assistant for dialogue summarization. Given the following dialogue between #Person1# and #Person2#, answer whether the given summary correctly summarizes the dialogue. Answer only 'Yes' or 'No'.
Example 1:
Dialogue: {dialogue}
Summary:{summary}
Answer:{answer}
Example 2:
...

**Prompt - Summarization - CoT:**
You are a helpful assistant for dialogue summarization. Given the following dialogue between #Person1# and #Person2#, answer whether the given summary correctly summarizes the dialogue. Let's work this out in a step-by-step way to be sure that we have the right answer. Then provide your final answer within the tags, <Answer>Yes/No</Answer>.
Example 1:
Dialogue:{dialogue}
Summary:{summary}
Answer:{answer}
Example 2:
...

**Prompt - Summarization + Commonsense Knowledge:**
You are a helpful assistant for dialogue summarization. Given the following dialogue between #Person1# and #Person2# and a relevant knowledge, answer whether the given summary correctly summarizes the dialogue. Answer only 'Yes' or 'No'.
Example 1:
Dialogue: {dialogue}
Summary:{summary}
Knowledge:{knowledge}
Answer:{answer}
Example 2:
...

**Prompt - Summarization - CoT Example:**
Dialogue:
#Person1#: I'm going to New York for the first time, but I don't have a tour guide. Can you give me any suggestions?
#Person2#: There's a service called 'A friend in New York'. It's a personal tour guide service.
#Person1#: That's interesting. What does it do?
#Person2#: You give them your information by answering a questionnaire and they will create a perfect trip for you according to your budget.
#Person1#: Good. Where can I get the questionnaire?
#Person2#: You can easily download it from their website.
#Person1#: That's helpful! Thanks!
Summary:
#Person1# is going to New York for the first time. #Person2# suggests #Person1# use a personal tour guide service even though they won't know how to put together #Person1#'s trip plan.
Answer:
Step 1: Identify the main points in the dialogue.
- #Person1# is going to New York for the first time and needs suggestions. - #Person2# suggests 'A friend in New York' service. - The service creates a perfect trip based on a questionnaire. - The questionnaire can be downloaded from their website.
Step 2: Compare the summary with the main points.
- The summary correctly mentions that #Person1# is going to New York for the first time. - The summary mentions the personal tour guide service, but it incorrectly states that they won't know how to put together #Person1#'s trip plan because according the dialogue, the service can create a perfect trip based on the questionnaire.
Final Answer: <Answer>No</Answer>

The template for CoT+Commonsense Knowledge is similar. We use the following example as a one-shot sample for CoT prompt of this task:

## D   Human Evaluation

In this section, we provide more details on the human evaluation results. As we allow human evaluators to discuss cases of disagreement, the number of resolutions and the human performance before and after the discussion are of interest as well. In Table 9, we report the percentage of resolved disagreements per task and the human results before and after discussion compared to the best-performant model which is GPT-4. Performance numbers for human scores before discussion are calculated by treating each annotator as a different prediction for each example and computing the performance over all predictions. If there is a clash and one annotator is correct and the other is not, then that example would receive a human score of $0.5$ for accuracy. After discussion, the annotators agree on the final prediction, so the agreed-upon label is the prediction. As we can see from the table, the human performance is already high before the disagreements are resolved, with GPT-4 only exceeding the individual human performance on stance classification. We note, however, that this is one of the tasks with the most initial disagreements. When discussed between both evaluators, however, they settle on labels and outperform GPT-4.

**Analysis of disagreements**   We observed different reasons for human disagreements, but the most common causes were: (i) instances requiring subtle factual knowledge that annotators may not recognize (see disagreement example below for Stance Classification), (ii) ambiguity and open-endedness (see disagreement example below for Safety Detection)

> **Disagreement Example - Stance:**
> *Belief*: Entrapment is bad if done properly.
> *Argument*: The bad parts of entrapment can be eliminated if the criminal can be given the opportunity to refuse.
> *Answer:* Yes (supporting)

Along with making commonsense inferences, the above example requires subtle factual knowledge about entrapment, namely that entrapment requires the criminal to not be aware that they are being entrapped. As a result, if the criminal is given the opportunity to refuse, then commonsense tells us that the entrapment is no longer being done properly since the criminal is aware. According to the be-

lief statement, if the entrapment is no longer being done properly, then it is no longer bad (contrapositive of the belief statement). This is the statement made by the argument, which is why the belief is supported by the argument. Without the subtle factual knowledge about entrapment, however, this example is challenging for human raters.

> **Disagreement Example - Safety:**
> *Scenario*: If your pets don't get along well
> *Action*: let them play in a large room together
> *Answer:* Yes (safe)

The above example is ambiguous because the action could result in both a safe and unsafe scenario. The action statement specifies that the room is large, hinting that it provides enough space for the pets to be in the same room and that playing together may let them grow closer (a common way of introducing pets to each other). However, "pets don't get along well" could also mean they attack each other and fight, and putting them in any room together might lead to an unsafe situation, regardless of their size. Without contextual information to clarify, the example remains ambiguous to human readers.

**Statistical Significance**   We perform a Binomial Sign test on GPT-4 and Human evaluation results as our sample size is small. We find that for Machine Translation (en-ru, en-de and en-fr), Intent detection, Safety detection and Stance Classification tasks, results are statistically significant with a level of 0.01, for Dialogue and Summarization tasks with a level of 0.05. Only for Machine Translation (zh-en) task, we obtain a high p-value (0.57).

## E   Additional Results

In this section, we provide the results of further experiments we did with varying hyperparameters. In Tables 10, 7 and 8, we report the averaged results of three runs where the temperature parameter is set to 0.3 respectively for all baselines, comparison over commonsense knowledge across tasks and with or without oracle knowledge. In Table 11, we report the results of GPT-4 with varying temperature values compared to human performance. In Table 12, we report the results of GPT-4 with chain-of-thought reasoning in various scenarios, including with self-consistency decoding (Wang et al., 2023). Overall, in none of the scenarios we observe a particularly different performance of the

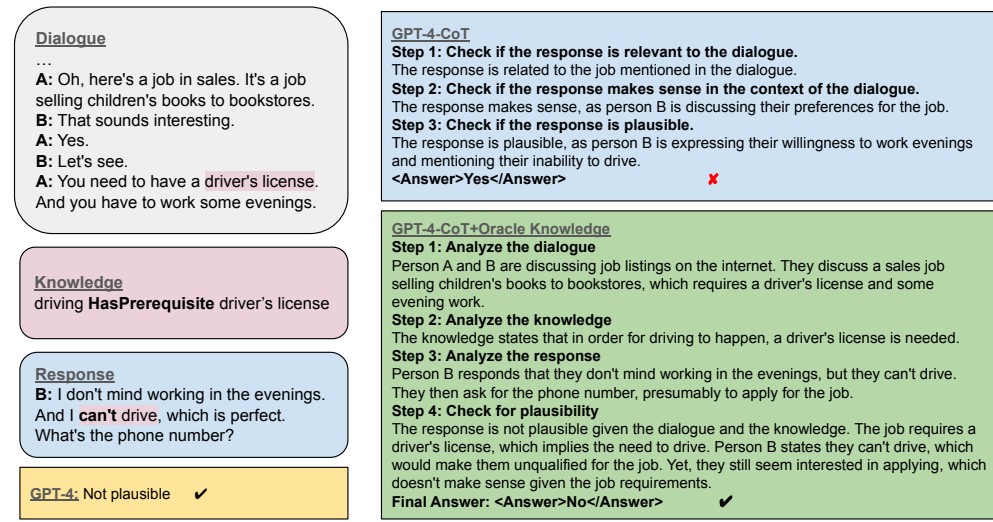

Figure 5: **Answers generated by GPT-4 in various settings for an example from the Dialogue task.** While chain-of-thought reasoning results in an incorrect answer, addition of the commonsense knowledge guides the model toward the correct answer.

models than what is reported in the main results in Table 2.

## Commonsense Knowledge Annotation

### Open-domain Dialogue

**Acceptance and Privacy Policies (click to expand/collapse)**

**Instructions (click to expand/collapse)**

*(WARNING: This HIT may contain adult content. Worker discretion is advised.)*
Thanks for participating in this HIT!

Given a **short dialogue** between two or more people, you are asked to **identify** 2 phrases from the dialogue and the *implicit* commonsense knowledge between these phrases.

**Commonsense knowledge** consists of *implicit* and *commonly known* facts about the everyday life such as *"Lemons are sour"* or *"If someone is hungry, then they would like to eat."*. It is a type of knowledge that nearly **everyone knows** intuitively and considers obvious, but is typically *implicit* which means it is rarely stated explicitly in text. For example, encyclopedic facts such as *"Ottawa is the capital of Canada"* are not considered commonsense knowledge because this information can be explicitly found online (e.g. Wikipedia) and not necessarily everyone knows it.

An example of a dialogue could be the following:

---

**Dialogue**

1. **#Person1#:** Hi, Sam, can you help me this weekend? I need help moving a new sofa into my house.
2. **#Person2#:** Hey, Jennifer, no problem. I'm free this weekend and my truck is great for moving stuff. Where did you get the sofa?
3. **#Person1#:** My friend Jack is moving next week, but his new apartment is very small. So **he is giving me his sofa.**
4. **#Person2#:** It's good that **your place is large enough to fit the sofa**. Where will you put it?

---

There are 4 **turns** in this dialogue and they alternate between #Person1# and #Person2#.

For example, in the above dialogue, #Person1# (Jennifer) needs to move a new sofa into his apartment and she got the sofa from her friend Jack whose new apartment didn't fit the new sofa. So, our commonsense knowledge allows us to infer that the #Person1#'s place is large enough to fit the sofa so she is getting it. Hence, we can connect the highlighted phrases **he is giving me his sofa** and **your place is large enough to fit the sofa** with an implicit relationship that the event in the first phrase *has the prerequisite* that the event in the second phrase happens (i.e. In order to get the sofa, your place should be large enough to fit it)

We can represent this type of commonsense knowledge with a relation called `HasPrerequisite` and write it as

`he is giving me his sofa` `HasPrerequisite` `your place is large enough to fit the sofa`

Since there could be many types of relationships in the context, to make the task easier, you will be provided with a list of knowledge relations and your task is to **identify 2 phrases** from the context that can be implicitly connected with one of the relations provided. If none of the relations provided describes the commonsense knowledge you have found, then you will be given an option to specify a **custom relationship**.

**Note that the connecting knowledge relationship for a correct commonsense knowledge must be commonly known and implicit (not explicitly stated in the context)**

Figure 6: Mturk Instructions template for Dialogue CKA stage

| Name | Description |
|------|-------------|
| **Attributional Relations** | |
| HasProperty | A has B as a property; A can be described as B. |
| CapableOf | Something that A can typically do is B. |
| HasA | B belongs to A, either as an inherent part or due to a social construct of possession. |
| HasSubEvent | A and B are events, and B happens as a subevent of A. |
| IsA | A is a subtype or a specific instance of B; every A is a B. |
| MannerOf | A is a specific way to do B. Similar to "Is A", but for verbs. |
| DependsOn | A depends on B. |
| CreatedBy | A is created by B. |
| **Physical/Spatial Relations** | |
| UsedFor | A is used for B. The purpose of A is B. |
| PartOf | A is part of B. |
| MadeOf | A is made up of B. |
| AtLocation | A happens at location B, or B is a typical location for A. |
| LocatedNear | A and B are typically found near each other. |
| **Temporal Relations** | |
| IsAfter | A happens after B. |
| IsBefore | A happense before B. |
| HappensIn | A happens during B. |
| IsSimultaneous | A and B happens at the same time. |
| HasPrerequisite | In order for A to happen, B needs to happen. |
| **Causal Relations** | |
| Causes | A causes B to happen. |
| Implies | A implies B. |
| HinderedBy | A is less likely to happen because of B. |
| **Social Relations** | |
| xIntent | Person in event A intends to do B. |
| xReact | Person in event A reacts as in B. |
| xNeed | Person in event A needs to do B before doing A. |
| xWant | Person in event A wants to do B. |
| xEffect | Event A will have the effect B on the Person in event A. |
| oReact | Others will react to event A as B. |
| oWant | Others will want to do B for A. |
| oEffect | Event A will have effect B on others. |
| MotivatedByGoal | Someone does A because they want result B. |
| **Comparative Relations** | |
| Antonym | A and B are opposites in some relevant way. |
| Synonym | A and B have very similar meanings. |
| SimilarTo | A is similar to B. |
| DistinctFrom | Something that is A is not B. |
| RelatedTo | A is related to B. |
| DefinedAs | A is defined as B. |

Table 5: List of commonsense relations

| Task | Context | Target | CKA/CKV | WSG/WSV | # Contexts | # Examples |
|------|---------|--------|---------|---------|-----------|-----------|
| Dialogue | dialogue | response | - | ✓ | 1,169 | 3,548 |
| Dialogue Summarization | dialogue | summary | ✓ | ✓ | 453 | 1,805 |
| Machine Translation (zh-en) | sentence | translation | ✓ | - | 600 | 1200 |
| Machine Translation (en-de) | sentence | translation | ✓ | - | 500 | 1000 |
| Machine Translation (en-fr) | sentence | translation | ✓ | - | 500 | 1000 |
| Machine Translation (en-ru) | sentence | translation | ✓ | - | 500 | 1000 |
| Intent Detection | news headline | intent | ✓ | ✓ | 589 | 2,440 |
| Stance Classification | belief/argument | argument/belief | - | ✓ | 397 | 1,722 |
| Safety Detection | scenario | action | ✓ | ✓ | 366 | 2,826 |
| Total | | | | | 5,074 | 16,541 |

Table 6: Overview of the benchmark. CKA/CKV stands for Commonsense Knowledge Annotation and Validation stages, WSG/WSV stands for Winograd Schema Generation and Validation stages, respectively. Stages are skipped (-) for tasks that already have the necessary data available.

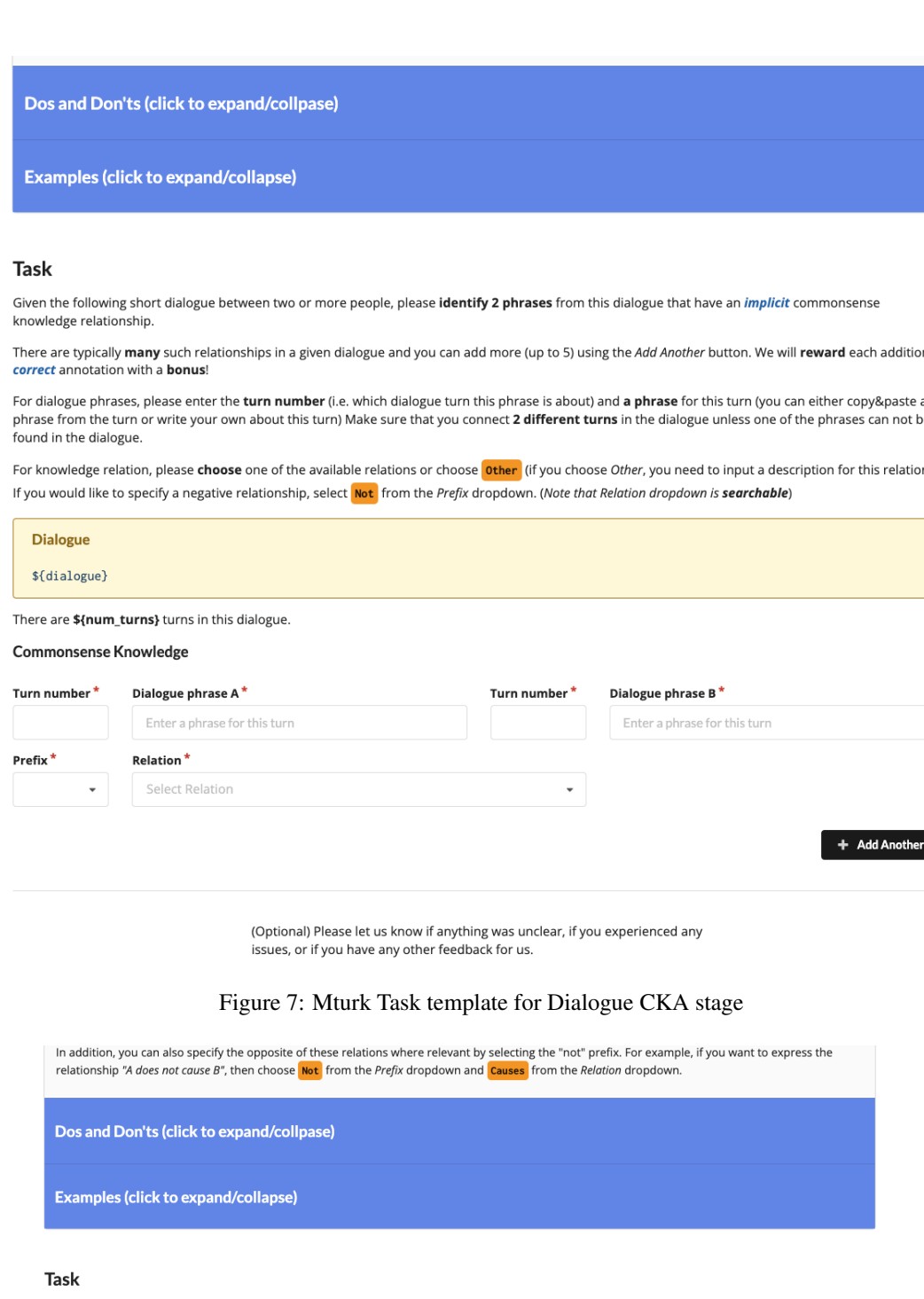

Figure 7: Mturk Task template for Dialogue CKA stage

Figure 8: Mturk Instructions template for Dialogue CKV stage

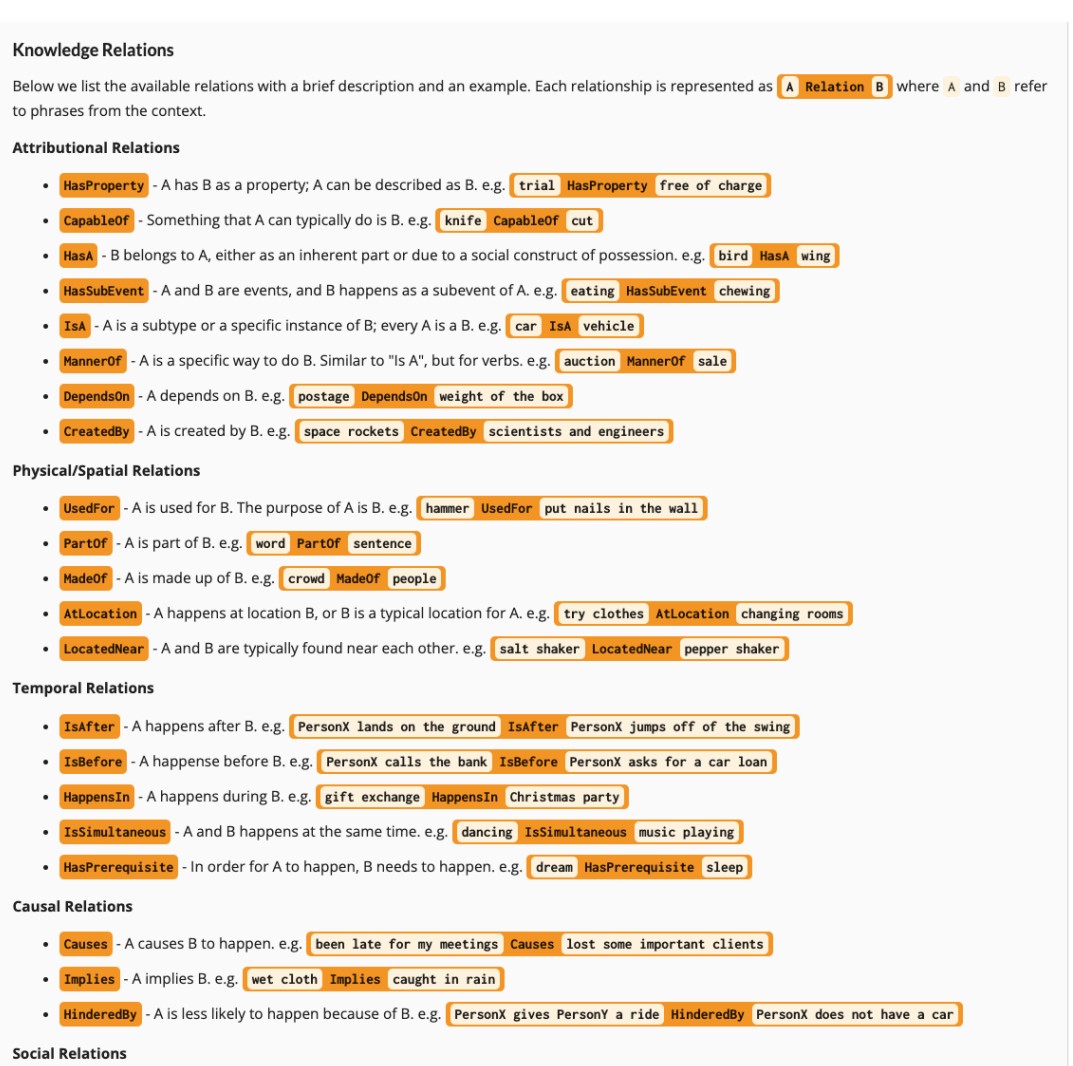

Figure 9: Mturk Knowledge Relations Section for Dialogue CKA stage

# Contextually Implausible Dialogue Turn Generation

**Acceptance and Privacy Policies (click to expand/collapse)**

## Instructions (click to expand/collapse)

*(WARNING: This HIT may contain adult content. Worker discretion is advised.)*
Thanks for participating in this HIT!

Given a **short dialogue** between two or more people, a **plausible final turn** of the dialogue and a **commonsense knowledge** connecting the dialogue and the final turn, you are asked to generate a **contextually implausible final turn** that satisfies the following conditions:

1. *minimally* differs from the original (plausible) final turn
2. *violates* the given commonsense knowledge
3. is *contextually implausible* final turn for the dialogue (something that wouldn't have followed the given dialogue, however, as a sentence on its own could be plausible!)
4. is still *relevant* for the given dialogue (something that is related to topics mentioned in the dialogue)

**Commonsense knowledge** consists of *implicit* and *commonly known* facts about the everyday life such as *"Lemons are sour"* or *"If someone is hungry, then they would like to eat."*. It is a type of knowledge that nearly **everyone knows** intuitively and considers obvious, but is typically *implicit* which means it is rarely stated explicitly in text. For example, encyclopedic facts such as *"Ottawa is the capital of Canada"* are not considered commonsense knowledge because this information can be explicitly found online (e.g. Wikipedia) and not necessarily everyone knows it.

An example of a dialogue could be:

---

**Dialogue**

1. **#Person1#:** Hi, Sam, can you help me this weekend? I need help moving a new sofa into my house.
2. **#Person2#:** Hey, Jennifer, no problem. I'm free this weekend and my truck is great for moving stuff. Where did you get the sofa?
3. **#Person1#:** My friend Jack is moving next week, but his new apartment is very small. So he is giving me his sofa.

---

There are 3 turns in this dialogue.

An example of a turn that could follow this dialogue:

---

**Plausible Final Turn**

4. **#Person2#:** It's good that your place is large enough to fit the sofa. Where will you put it?

---

An example of a commonsense knowledge connecting this dialogue and the final turn could be:

Figure 10: Mturk Instructions template for Dialogue WSG stage

**Task**

Given the following **short dialogue** between two or more people, *a plausible final turn* for the dialogue, and a list of **commonsense knowledge** instances connecting the dialogue and the given turn, please select one or more commonsense knowledge instances and write a *contextually implausible final turn* that satisfies the following conditions:

1. *minimally* differs from the original (plausible) final turn

2. *violates* the chosen commonsense knowledge

3. is *contextually implausible* final turn for the dialogue (something that wouldn't have followed the given dialogue, however, as a sentence on its own could be plausible!)

4. is still *relevant* for the given dialogue (something that is related to topics mentioned in the dialogue)

**Please, follow the guidelines** mentioned above closely to make sure your suggestion satisfies all the conditions.

**Please, note that** simply replacing words/phrases with their antonyms or negating sentences do **NOT** necessarily make them *contextually implausible.* Contextually implausible final turn is a turn that does not clearly make sense within the full context of the dialogue. Please, see the instructions for an example.

**Please, also avoid** turns that are implausible even *independent* of the given dialogue.

There are typically multiple ways to generate this contextually implausible final turn, so feel free to add 2 more alternatives by using the *Add Another* button. Note that **you will be paid a bonus for each additional contextual implausible final turn that satisfies our conditions for this task.**

*Note that you are free to modify* *any* *part of the final turn, not only the highlighted parts as long as it satisfies our conditions for this task.*

---

**Dialogue**

${dialogue}

---

**Plausible Final Turn**

${final_turn}

---

**Commonsense Knowledge**

*For each annotation, choose one or more commonsense knowledge instances and write a new final turn that satisfies the conditions given above.*

---

**Commonsense Knowledge** *

| Select Commonsense Knowledge | ▾ |

**Contextually Implausible Final Turn 1** *

${final_turn_prefix}

| Write a minimally different contextually implausible final turn that violates the chosen commonsense knowledge. |

**+ Add Another**

Figure 11: Mturk Task template for Dialogue WSG stage

## Task

Given the following **short dialogue** between 2 or more people, a *plausible* final turn, a list of **commonsense knowledge** instances connecting the final turn with the dialogue and a list of **potentially implausible** final turns, you are asked to **choose** the final turn(s) that satisfy the following conditions:

1. *minimally* differs from the original plausible one

2. *violates* the given commonsense knowledge

3. is a *contextually implausible* final turn for the dialogue (something that wouldn't have followed the given dialogue, but as a sentence on its own, could be plausible)

4. is still *relevant* for the given dialogue (something that is related to topics mentioned in the dialogue)

If **none of the options** satisfy the conditions, then tick the option *None of the above* and provide the implausible turn that you think satisfies the conditions (you can enter up to 3 and each additional correct annotation will be rewarded with a bonus)

---

**Dialogue**

${dialogue}

---

**Plausible Final Turn**

${final_turn}

---

**Commonsense Knowledge**

**(Potentially Implausible) Final Turns**

**Which of the following final turns satisfy the conditions above?** (Select all that apply)

☐ **None of the above.**

---

Figure 12: Mturk Instructions template for Dialogue WSV stage

# Commonsense Knowledge Annotation

## Qualification Round

### Acceptance and Privacy Policies (click to expand/collapse)

### Instructions (click to expand/collapse)

*(WARNING: This HIT may contain adult content. Worker discretion is advised.)*

Thanks for participating in this HIT!

*Note: Please, avoid using multiple accounts for this HIT. We will reject the work if we detect such cases of misuse!*

**This HIT is our Qualification Round to build an exclusive pool of annotators who will get to participate in a several-stage data collection process. We intend to release hundreds of HITs in this project and compensate adequately with a bonus for each additional annotation per HIT.**

**Commonsense knowledge** consists of *implicit* and *commonly known* facts about the everyday life such as *"Lemons are sour"* or *"If someone is hungry, then they would like to eat."*. It is a type of knowledge that nearly **everyone knows** intuitively and considers obvious, but is typically *implicit* which means it is rarely stated explicitly in text. For example, encyclopedic facts such as *"Ottawa is the capital of Canada"* are not considered commonsense knowledge because this information can be explicitly found online (e.g. Wikipedia) and not necessarily everyone knows it.

We are running a large-scale project to collect high-quality commonsense knowledge examples related to various real-life situations described in the form of dialogues, summaries or simply pairs of sentences.

In order to accomplish this, we are looking for creative people like you to help us identify such implicit knowledge given some text.

This HIT has **2 independent parts** that will be used to test your ability to perform the required tasks in our project.

- The first part consists of **multiple-choice questions** about identifying correct commonsense knowledge from a list of options.
- The second part contains **open-ended questions** which will require you to come up with a commonsense knowledge about a given context. **Future HITs that we will release will all resemble this second part.**

#### Part 1 (Multiple-choice) instructions

In this part, you will be given some context in various formats (e.g dialogue, sentences) and several options of knowledge instances and asked to identify those that are *correct implicit* commonsense knowledge examples about the given context. This part has 3 tasks.

#### Task 1 instructions

In this task, the context will be given in the form of a **dialogue** and asked to identify correct commonsense knowledge connecting **different turns** of the dialogue.

An example of a dialogue could be the following:

#### Dialogue

Figure 13: Mturk Instructions template for Qualification Stage

**Part 1 (Multiple-choice questions)**

In this part, you will be given some context in various formats (e.g dialogue, sentences) and several options of knowledge instances and asked to identify those that are *correct implicit* commonsense knowledge examples about the given context.

**Task 1**

In this task, you are given **short dialogues** between two or more people, and a list of **knowledge instances** connecting different turns in this dialogue. You need to **select** ones that express an *implicit* commonsense knowledge about the given dialogue.

**Question 1**

> **Dialogue**
>
> ${mcq_odd_q1_dialogue}

**Which of the following knowledge instances are correct implicit commonsense knowledge about this dialogue?** (Select all that apply)

☐ ${mcq_odd_q1_opt1_head} ${mcq_odd_q1_opt1_rel} ${mcq_odd_q1_opt1_tail}

☐ ${mcq_odd_q1_opt2_head} ${mcq_odd_q1_opt2_rel} ${mcq_odd_q1_opt2_tail}

☐ ${mcq_odd_q1_opt3_head} ${mcq_odd_q1_opt3_rel} ${mcq_odd_q1_opt3_tail}

☐ ${mcq_odd_q1_opt4_head} ${mcq_odd_q1_opt4_rel} ${mcq_odd_q1_opt4_tail}

**Question 2**

> **Dialogue**
>
> ${mcq_odd_q2_dialogue}

**Which of the following knowledge instances are correct implicit commonsense knowledge about this dialogue?** (Select all that apply)

☐ ${mcq_odd_q2_opt1_head} ${mcq_odd_q2_opt1_rel} ${mcq_odd_q2_opt1_tail}

☐ ${mcq_odd_q2_opt2_head} ${mcq_odd_q2_opt2_rel} ${mcq_odd_q2_opt2_tail}

Figure 14: Mturk Task template for Qualification Stage Part 1 (MCQ)

**Part 2 (Open-ended questions)**

In this part, you will be given a context and asked to write an *implicit* commonsense knowledge about this context.

**Question**

Given the following short dialogue between two or more people, please **identify 2 phrases** from this dialogue that have an *implicit* commonsense knowledge relationship.

There are typically **many** such relationships in a given dialogue and you can add more (up to 3) using the *Add Another* button. We will **reward** each additional *correct* annotation with a **bonus** and **qualify** you for our project!

For dialogue phrases, please enter the **turn number** (i.e. which dialogue turn this phrase is about) and **a phrase** for this turn (you can either copy&paste a phrase from the turn or write your own about this turn) Make sure that you connect **2 different turns** in the dialogue.

For knowledge relation, please **choose** one of the available relations or choose Other and input a description for this relation. If you would like to specify a negative relationship, select Not from the *Prefix* dropdown. (*Note that Relation dropdown is* **searchable**)

> **Dialogue**
>
> ${oq_odd_q1_dialogue}

There are **${oq_odd_q1_num_turns}** turns in this dialogue.

**Commonsense Knowledge**

| Turn number* | Dialogue phrase A* | Turn number* | Dialogue phrase B* |
|---|---|---|---|
| | Enter a phrase for this turn | | Enter a phrase for this turn |

| Prefix* | Relation* |
|---|---|
| ▼ | Select Relation ▼ |

**+ Add Another**

(Optional) Please let us know if anything was unclear, if you experienced any issues, or if you have any other feedback for us.

Figure 15: Mturk Task template for Qualification Stage Part 2 (Open-ended)

| Model | CK Dimensions | | | | | |
|---|---|---|---|---|---|---|
| | Attribution | Physical | Temporal | Causal | Social | Comparison |
| **Flan-Alpaca♣** | 67.4 | 69.4 | 66.1 | 67.6 | 70.4 | 67.5 |
| **Flan-T5-11B♣** | 75.5 | 76.6 | 76.5 | 75.9 | 77.1 | 79.2 |
| **LLaMa-33B♣** | 42.2 | 42.4 | 44.2 | 42.0 | 42.4 | 43.5 |
| **Stable-Vicuna♣** | 55.0 | 56.9 | 59.6 | 55.3 | 56.1 | 56.1 |
| **BloomZ-7B** | 56.1 | 56.7 | 54.9 | 54.6 | 55.1 | 56.2 |
| **PaLM-1-540B** | 48.0 | 48.4 | 49.8 | 48.2 | 49.4 | 49.4 |
| **GPT-3.5** | 64.5 | 64.3 | 63.5 | 64.1 | 65.9 | 70.5 |
| **GPT-4** | 74.3 | 72.5 | 70.9 | 73.5 | 73.0 | 72.4 |
| **GPT-4-CoT** | 73.4 | 71.5 | 68.8 | 71.7 | 73.6 | 72.9 |

Table 7: **Macro-F1** scores averaged across commonsense knowledge dimensions (♣all tasks except for MT.). Temperature is set to 0.3 and all results are averaged over three runs with different seeds.

| Model | Oracle Knowledge | No Knowledge |
|---|---|---|
| **Flan-T5-11B♣** | 77.0 / 46.2 | 75.0 / 41.1 |
| **BloomZ-7B** | 52.3 / 13.6 | 55.5 / 15.9 |
| **GPT-4** | 74.5 / 47.5 | 72.9 / 45.4 |
| **GPT-4-CoT** | 76.5 / 52.8 | 72.2 / 46.3 |

Table 8: **Macro-F1 / Situational Accuracy** scores averaged over all tasks (♣all tasks except MT), with and without providing commonsense knowledge in the prompt. Temperature is set to 0.3 and all results are averaged over three runs with different seeds.

| Models | MT | | | | DG | DS | SC | SD | ID | CROW Score (-MT) | CROW Score |
| --- | --- | --- | --- | --- | --- | --- | --- | --- | --- | --- | --- |
| | Zh-En | En-Fr | En-De | En-Ru | | | | | | | |
| **GPT-4** | 75.9 / 57.9 | 54.5 / 21.5 | 54.4 / 20.5 | 54.1 / 19.7 | 72.4 / 46.5 | 89.6 / 75.3 | 79.6 / 54.7 | 89.7 / 51.9 | 84.0 / 57.2 | 83.1 / 57.1 | 72.7 / 45.0 |
| **Human (before discussion)** | 87.4 / 78.0 | 75.5 / 75.5 | 85.3 / 75.0 | 84.4 / 76.0 | 86.5 / 86.0 | 97.3 / 91.1 | 83.5 / 56.5 | 90.1 / 75.8 | 92.4 / 73.1 | 90.0 / 76.5 | 86.9 / 76.3 |
| **Human (after discussion)** | 87.9 / 78.0 | 83.0 / 82.9 | 89.9 / 82.0 | 89.9 / 86.0 | 87.0 / 86.9 | 98.9 / 96.4 | 88.1 / 69.6 | 97.8 / 93.9 | 93.9 / 80.7 | 93.1 / 85.5 | 90.7 / 84.0 |
| **Resolutions** | 7% | 18% | 12% | 15% | 3% | 5% | 18% | 19% | 9% | | |

Table 9: Human Evaluation results before and after discussion compared to GPT-4 and the percentage of resolved disagreements per task.

| Models | MT | | | | DG | DS | SC | SD | ID | CROW Score (-MT) | CROW Score |
| --- | --- | --- | --- | --- | --- | --- | --- | --- | --- | --- | --- |
| | Zh-En | En-Fr | En-De | En-Ru | | | | | | | |
| **Majority** | 33.3 / 0.0 | 33.3 / 0.0 | 33.3 / 0.0 | 33.3 / 0.0 | 40.1 / 0.0 | 42.8 / 0.0 | 33.6 / 0.0 | 36.5 / 0.0 | 41.3 / 0.0 | 38.9 / 0.0 | 36.4 / 0.0 |
| **Random** | 49.9 / 25.4 | 50.0 / 24.7 | 50.3 / 24.9 | 49.0 / 23.8 | 48.1 / 14.4 | 46.4 / 9.9 | 50.6 / 6.9 | 49.8 / 0.6 | 48.3 / 10.2 | 48.6 / 8.4 | 49.1 / 15.6 |
| **LLaMa-7B** | 33.3 / 0.0 | – | – | – | 41.5 / 1.3 | 44.3 / 1.4 | 33.7 / 0.0 | 29.9 / 0.0 | 41.3 / 0.0 | 38.1 / 0.5 | 37.3 / 0.5 |
| **LLaMa-13B** | 46.0 / 13.8 | – | – | – | 50.2 / 10.9 | 45.1 / 2.1 | 34.4 / 0.7 | 30.6 / 0.1 | 43.8 / 1.8 | 40.8 / 3.1 | 41.7 / 4.9 |
| **LLaMa-33B** | 33.3 / 0.1 | – | – | – | 52.5 / 4.9 | 49.6 / 5.6 | 33.7 / 0.2 | 30.1 / 0.0 | 41.3 / 0.0 | 41.4 / 2.1 | 40.1 / 1.8 |
| **Flan-T5-11B** | 57.0 / 26.8 | – | – | – | 68.6 / 39.0 | 64.7 / 30.7 | 75.4 / 48.2 | 83.0 / 56.8 | 83.3 / 56.8 | 75.0 / 41.1 | 72.0 / 38.7 |
| **Alpaca** | 38.4 / 4.7 | – | – | – | 54.8 / 16.7 | 56.5 / 12.3 | 41.0 / 6.1 | 43.8 / 2.4 | 52.0 / 9.4 | 49.6 / 9.4 | 47.8 / 8.6 |
| **Flan-Alpaca** | 60.2 / 26.5 | – | – | – | 62.5 / 28.0 | 52.0 / 18.4 | 67.2 / 36.4 | 67.0 / 11.3 | 78.5 / 46.0 | 65.5 / 28.0 | 64.6 / 27.8 |
| **Vicuna** | 37.6 / 4.2 | – | – | – | 60.4 / 21.4 | 61.9 / 18.9 | 45.1 / 11.7 | 42.0 / 1.6 | 57.2 / 15.0 | 53.3 / 13.7 | 50.7 / 12.1 |
| **Stable-Vicuna** | 60.5 / 30.9 | – | – | – | 52.0 / 11.9 | 38.7 / 7.2 | 51.1 / 17.8 | 68.6 / 14.3 | 63.9 / 23.6 | 54.9 / 15.0 | 55.8 / 17.6 |
| **mT0** | 55.1 / 20.2 | 37.7 / 3.3 | 35.6 / 1.5 | 33.9 / 0.3 | 44.8 / 4.8 | 64.0 / 23.7 | 55.2 / 14.6 | 50.8 / 2.8 | 49.5 / 6.2 | 52.8 / 10.4 | 47.4 / 8.6 |
| **BloomZ-7B** | 55.1 / 19.8 | 47.0 / 15.0 | 49.6 / 18.0 | 49.2 / 17.9 | 52.1 / 11.1 | 56.5 / 15.1 | 57.7 / 16.2 | 68.3 / 8.5 | 64.3 / 21.8 | 59.8 / 14.5 | 55.5 / 15.9 |
| **PaLM 1** | 33.8 / 0.6 | 34.2 / 1.1 | 34.1 / 0.9 | 33.4 / 0.3 | 63.6 / 26.8 | 62.6 / 23.0 | 51.6 / 15.4 | 56.6 / 7.9 | 63.5 / 21.9 | 59.6 / 19.0 | 48.2 / 10.9 |
| **GPT-3** | 66.7 / 39.4 | 48.5 / 16.1 | 49.2 / 17.2 | 48.1 / 12.1 | 67.1 / 36.8 | 68.6 / 32.6 | 69.2 / 38.2 | 85.6 / 39.5 | 76.4 / 42.0 | 73.4 / 37.8 | 64.2 / 29.7 |
| **GPT-4** | 75.6 / 56.6 | 54.5 / 22.1 | 54.6 / 20.3 | 53.8 / 20.5 | 72.0 / 45.6 | 90.5 / 77.3 | 81.5 / 57.3 | 89.2 / 50.2 | 84.7 / 59.0 | 83.6 / 57.9 | 72.9 / 45.4 |
| **GPT-4-CoT** | 71.2 / 51.1 | 64.3 / 41.9 | 57.5 / 34.3 | 56.9 / 29.2 | 55.2 / 23.1 | 89.0 / 71.7 | 83.6 / 60.6 | 88.2 / 47.0 | 84.2 / 57.8 | 80.0 / 52.0 | 72.2 / 46.3 |
| **Human***  | 87.9 / 78.0 | 83.0 / 82.9 | 89.9 / 82.0 | 89.9 / 86.0 | 87.0 / 86.9 | 98.9 / 96.4 | 88.1 / 69.6 | 97.8 / 93.9 | 93.9 / 80.7 | 93.1 / 85.5 | 90.7 / 84.0 |

Table 10: **Macro-F1 / Situational Accuracy** (*i.e.*, results aggregated per *context* instead of per *sample*) for all examined models across CROW tasks. All model results are averaged over three runs with different seeds. Temperature is set to 0.3 for all runs. *Due to the cost of expert evaluation, our **Human** study is only evaluated on 100 instances per task.

| Models | MT | | | | DG | DS | SC | SD | ID | CROW Score (-MT) | CROW Score |
| --- | --- | --- | --- | --- | --- | --- | --- | --- | --- | --- | --- |
| | Zh-En | En-Fr | En-De | En-Ru | | | | | | | |
| **GPT-4 (temp=0.0)** | 75.9 / 57.9 | 54.5 / 21.5 | 54.4 / 20.5 | 54.1 / 19.7 | 72.4 / 46.5 | 89.6 / 75.3 | 79.6 / 54.7 | 89.7 / 51.9 | 84.0 / 57.2 | 83.1 / 57.1 | 72.7 / 45.0 |
| **GPT-4 (temp=0.1)** | 76.4 / 58.1 | 53.3 / 20.7 | 54.0 / 19.9 | 53.7 / 20.3 | 72.5 / 46.0 | 89.5 / 75.3 | 79.9 / 54.9 | 89.4 / 50.8 | 83.5 / 56.1 | 83.0 / 56.7 | 72.5 / 44.7 |
| **GPT-4 (temp=0.3)** | 75.6 / 56.6 | 54.5 / 22.1 | 54.6 / 20.3 | 53.8 / 20.5 | 72.0 / 45.6 | 90.5 / 77.3 | 81.5 / 57.3 | 89.2 / 50.2 | 84.7 / 59.0 | 83.6 / 57.9 | 72.9 / 45.4 |
| **Human***  | 87.9 / 78.0 | 83.0 / 82.9 | 89.9 / 82.0 | 89.9 / 86.0 | 87.0 / 86.9 | 98.9 / 96.4 | 88.1 / 69.6 | 97.8 / 93.9 | 93.9 / 80.7 | 93.1 / 85.5 | 90.7 / 84.0 |

Table 11: **Macro-F1 / Situational Accuracy** (*i.e.*, results aggregated per *context* instead of per *sample*) for GPT-4 across CROW tasks with varying temperature values.

| Models | MT | | | | DG | DS | SC | SD | ID | CROW Score (-MT) | CROW Score |
| --- | --- | --- | --- | --- | --- | --- | --- | --- | --- | --- | --- |
| | Zh-En | En-Fr | En-De | En-Ru | | | | | | | |
| **GPT-4-CoT (temp=0.0)** | 71.6 / 52.2 | 64.7 / 42.6 | 57.1 / 34.2 | 57.3 / 30.0 | 55.3 / 22.8 | 88.6 / 70.6 | 84.3 / 60.7 | 87.8 / 47.3 | 84.0 / 57.0 | 80.0 / 51.7 | 72.3 / 46.4 |
| **GPT-4-CoT (temp=0.3, average)** | 71.3 / 51.1 | 64.4 / 42.0 | 57.4 / 34.1 | 56.6 / 28.9 | 55.2 / 22.8 | 88.8 / 71.3 | 83.7 / 60.8 | 88.1 / 47.5 | 83.9 / 57.3 | 80.0 / 51.9 | 72.2 / 46.2 |
| **GPT-4-CoT (temp=0.3, majority)** | 71.5 / 51.3 | 64.2 / 42.0 | 58.1 / 35.2 | 56.0 / 27.8 | 55.1 / 23.0 | 89.6 / 73.3 | 83.1 / 59.4 | 87.9 / 45.6 | 84.2 / 57.9 | 80.0 / 51.9 | 72.2 / 46.2 |
| **Human***  | 87.9 / 78.0 | 83.0 / 82.9 | 89.9 / 82.0 | 89.9 / 86.0 | 87.0 / 86.9 | 98.9 / 96.4 | 88.1 / 69.6 | 97.8 / 93.9 | 93.9 / 80.7 | 93.1 / 85.5 | 90.7 / 84.0 |

Table 12: **Macro-F1 / Situational Accuracy** (*i.e.*, results aggregated per *context* instead of per *sample*) for GPT-4 with CoT across CROW tasks in different scenarios. In the 'average' scenario, an average of five experiment results are reported. In the 'majority' scenario, similar to (Wang et al., 2023), results based on the majority answer from five experiments are reported.