# OpenReview forum: "CRoW: Benchmarking Commonsense Reasoning in Real-World Tasks"
_EMNLP/2023/Conference — EMNLP 2023 Main_

### Official Review · Reviewer_H1md · 2023-08-01

**Soundness:** 3

**Excitement:**

5: Transformative: This paper is likely to change its subfield or computational linguistics broadly. It should be considered for a best paper award. This paper changes the current understanding of some phenomenon, shows a widely held practice to be erroneous in someway, enables a promising direction of research for a (broad or narrow) topic, or creates an exciting new technique.

**Missing References:**

ProPara (https://arxiv.org/abs/1805.06975) and TRIP (https://aclanthology.org/2021.findings-emnlp.422.pdf) may be relevant physical commonsense benchmarks to cover in Section 2.

There are some papers that that propose robustness metrics similar in spirit to your Situational Accuracy metric: “Group Consistency” in Ravichander et al. (2022; https://arxiv.org/pdf/2211.00295.pdf) and “Strict Coherence” in Storks and Chai (2021; https://aclanthology.org/2021.findings-emnlp.272.pdf) come to mind. It might not be necessary to cite them, but your work reminded me of them.

**Paper Topic And Main Contributions:**

Motivated by recent rapid advances on NLP benchmarks for commonsense reasoning made by large language models (LLMs), this work argues that the artificial scenarios provided in existing benchmarks for commonsense reasoning are not reflective of real-world NLP tasks. As such, they propose a new benchmark CRoW, which applies a novel data collection paradigm to generate Winograd schema-like rewrites of existing NLP task examples, where systems would be expected to judge whether a candidate response to one of these tasks is valid.

Their benchmark includes evaluations across 6 practical task settings (machine translation, open-domain dialogue, dialogue summarization, intent detection, stance classification, and safety detection, which cover 7 sub-domain categories of commonsense (temporal, causal, attribution, comparison, physical, social, and other). Upon evaluating 13 SOTA systems on CRoW, including GPT-4, the authors find that they fall short on real-world practical commonsense reasoning.

This paper presents a substantial data collection and evaluation effort, and the CRoW dataset is a useful and systematic tool that can be used by the community to more objectively evaluate NLP systems on commonsense reasoning. The methodology is remarkably thorough, and the paper is well written. However, some important details about model evaluation seem to be missing that would help me better understand the papers’ conclusions.

**Questions For The Authors:**

*Question A:* Did you try to use the LLMs in this study to generate valid responses for the language contexts? I wonder whether generating responses may somehow be easier or less confusing for LLMs than judging the validity of them (which may be subjective, especially in zero-shot settings).

*Question B:* Are the model evaluations all zero-shot? I couldn’t find this detail. If so, is it possible that the LLM doesn’t quite pick up the format of this task (which is still a bit unusual despite being more relevant to real-world tasks) without some demonstrations?

*Question C:* I was a bit confused by the metric choices in Tables 2 and 3. Why do we use macro-F1 in Table 2 but accuracy in Table 3? This makes it hard to compare the results across the two tables and understand the relationship there.

**Reasons To Accept:**

*Strength A:* The paper is well motivated and timely, calling into question current benchmarking practices in NLP and demonstrating that they’re insufficient to evaluate systems for common practical settings. The message of the paper is well supported, and the community would benefit from it: the reasoning ability of NLP systems only really counts when it comes to their practical real-world application.

*Strength B:* The data collection method and benchmark presented are a substantial and comprehensive effort, and can be beneficial to the community and used by others to more objectively evaluate and facilitate improvement of NLP systems on commonsense reasoning, thought of as one of the hardest challenges in AI.

*Strength C:* The data collection and evaluation procedures presented in the paper are quite thorough. The benchmark covers a comprehensive set of practical NLP tasks selected from papers going back to 2000, and a wide range of commonsense knowledge sub-domains based on prior work. Data is validated in multiple ways and often by multiple humans after every step of the collection process. Evaluation metrics include not only accuracy on end tasks, but a deeper situational accuracy which judges the completeness of systems’ commonsense reasoning for each individual language context used in the benchmark. 13 diverse and relevant language models are evaluated, and discussion and analysis of results provides a variety of insights about the characteristics of models and the datasets.

**Reasons To Reject:**

*Weakness A:* From what I can tell, the paper is missing important details about how LLMs are evaluated. While some prompt templates are given in the appendix, it’s actually not clear to me whether these were given to annotators or LLMs. It is unclear whether any demonstrations of tasks are given within prompts (i.e., for in-context learning), and if this is not the case, this raises questions about whether LLMs just have trouble picking up the format of the tasks rather than struggling with commonsense reasoning. It would also be important to know whether any randomness is involved in generating language, and how this is controlled.

*Weakness B:* The human evaluation is a bit small-scale. While differences between human results and system results seem wide enough for this to be conclusive, I would be curious whether the differences are statistically significant, e.g., using a McNemar test.

If the authors provide sufficient detail to address these possible issues, I will consider modifying my soundness score.

**Reproducibility:**

3: Could reproduce the results with some difficulty. The settings of parameters are underspecified or subjectively determined; the training/evaluation data are not widely available.

**Reviewer Confidence:**

4: Quite sure. I tried to check the important points carefully. It's unlikely, though conceivable, that I missed something that should affect my ratings.

**Typos Grammar Style And Presentation Improvements:**

L078: Aren’t there 7 dimensions of commonsense annotated? I think you missed “social” here and also in the Limitations section. You may want to double check that you’re consistent about the categories used.

L337: Small grammatical error in parentheses; can remove “the” to fix it.

L891: the reference here has some extra numbers inserted into it.

---

> ### Author Rebuttal · Authors · 2023-08-29
>
> We thank the reviewer for their thoughtful comments and helpful suggestions. We are happy that the reviewer finds our work quite exciting. We will take into account their suggestions in the camera-ready version of our paper.
>
> **Missing important details about LLM evaluation**
>
> We thank the reviewer for pointing out several missing important details about LLM evaluation from the paper that we will add in the camera-ready version for better clarity.
>
> *Prompt templates*: We used the same prompt templates for LLM and human evaluation for fair evaluation.
>
> *Demonstrations of tasks*:  LLMs were evaluated in one-shot fashion for all tasks. We chose this setting for simplicity (we have several tasks and several models to evaluate) while making sure the models pick up the format of the task.
>
> *Randomness in Generation*: We haven’t observed any particular randomness in generating language with the evaluated models. We used the following hyperparameters which we found to result in good performance.
>
> (1) GPT-4 model: max_tokens=40, temperature=0.3, top_p=1
>
> (2) Other oper-sourced models: max_tokens=1, temperature=0.5, top_k=20, top_p =0.95
>
> **Why were there only 2 annotators chosen for the human evaluation and what is the statistical significance of evaluation results?**
>
> We thank the reviewer for bringing up these important points about the reliability of our human evaluation results. First, our setup may only have two annotators per task, but these are expert annotators who are NLP researchers from our lab (importantly, they were not involved in the original data collection). As a result, the evaluators are more experienced, can clarify misunderstandings in the annotation guideline with us, and generally produce more careful annotations than crowd workers. Finally, following [1, 2], we intentionally allowed annotators to discuss in cases of disagreement to reduce variance and yield a robust upper bound for our task. In the camera-ready version, we will clarify these important points about our human study.
>
> Based on the reviewer’s suggestion, we performed the binomial sign test on GPT-4 and human evaluation results as our sample size is small. We find that for Machine Translation (en-ru, en-de and en-fr), Intent detection and Safety detection tasks, results are statistically significant with a level of 0.01, for Dialogue and Summarization tasks with a level of 0.1. For Stance detection and zh-en machine translation task, we find the p-values slightly high (0.16 and 0.45 respectively), however, this is also aligned with our overall results with respect to the performance of GPT-4 on these tasks and the observed narrow gap. We will add these results to the camera-ready.
>
> [1] Amidei, Jacopo et al. 2018 Rethinking the Agreement in Human Evaluation Tasks, ACL
>
> [2] Oortwijn et al., 2021 Interrater Disagreement Resolution: A Systematic Procedure to Reach Consensus in Annotation Tasks, ACL
>
> **Generating valid responses using LLMs instead of judging their validity**
>
> While evaluating generated responses would be an interesting avenue, we decided to use classification-based evaluation of minimal pairs as a starting point. Generating responses is open-ended, meaning we may not always immediately uncover the differences along the commonsense dimensions we are interested to explore, which would not fully test the robustness of these models to semantic and pragmatic subtleties. However, the reviewer points out an interesting direction for future work, and one we are actively exploring at the moment.
>
> **Why did we report accuracy in Table 3?**
>
> We report Accuracy on negative samples in Table 3. During our annotation pipeline, commonsense dimensions are linked explicitly to the negative samples (which are constructed using a provided commonsense violation). For positive examples, multiple pieces of commonsense may be annotated (from the CKA stage). As a result, there is no 1-1 correspondence between positive samples and a piece of commonsense knowledge, so we opted to analyze the performance of these models on the commonsense knowledge dimension for negative samples only and report the results in Table 3.
>
> For the sake of completeness, we ran the new CK analysis with positive samples included (where each positive sample may have more than one piece of commonsense knowledge associated with it) and report the results below (Macro-F1 reported):
>
> |             |  **Attribution** | **Physical** | **Temporal** | **Causal** | **Social** | **Comparison** |
> | ----------- | ------------ | -------- | -------- | ------ | ------ | ---------- |
> | Flan-Alpaca | 69.9         | 71.6     | 54.3     | 70.2   | 71.8   | 73.5       |
> | Flan-T5     | 76.8         | 77.8     | 63.1     | 78.1   | 79.3   | 80.0       |
> | LLaMa       | 46.5         | 45.9     | 37.4     | 45.7   | 46.1   | 45.3       |
> | Stable-Vicuna | 54.9       | 56.2     | 44.2     | 53.8   | 57.8   | 55.1       |
> | BloomZ-7B   | 54.3         | 54.9     | 46.5     | 53.7   | 52.7   | 52.9       |
> | PaLM        | 55.4         | 53.6     | 50.1     | 53.9   | 57.6   | 53.9       |
> | GPT-3       | 64.5         | 65.2     | 57.2     | 62.8   | 65.9   | 69.2       |
> | GPT-4       | 75.6         | 73.9     | 64.9     | 74.3   | 74.6   | 75.5       |
>
> We will include this result in the camera-ready version of our paper. As stated above, though, these results inflate the contribution of each positive sample to the final result (as they may be associated with multiple pieces of commonsense knowledge).

---

### Official Review · Reviewer_x28y · 2023-08-05

**Typos Grammar Style And Presentation Improvements:** 1. To make the comparison of model pe…
**Soundness:** 3

**Excitement:**

4: Strong: This paper deepens the understanding of some phenomenon or lowers the barriers to an existing research direction.

**Paper Topic And Main Contributions:**

The paper presents CROW, a benchmark for evaluating commonsense reasoning in real-world NLP tasks. The authors propose a multi-stage data collection pipeline based on commonsense-violating perturbations following Winograd schemas. They apply this pipeline to construct CROW, comprising six real-world NLP tasks. The authors evaluate the performance of various language models on CROW and find that the models still fall far below human performance regarding commonsense reasoning. They also analyze the models' performance across different dimensions of commonsense knowledge.

**Reasons To Accept:**

1. The paper proposes a novel approach by combining commonsense reasoning with real-world NLP tasks, effectively bridging the gap between the commonsense reasoning domain and practical NLP applications. Unlike previous studies that mainly focused on probing commonsense reasoning ability in isolation, this work provides valuable insights into its practical implications.
2. This paper comprehensively covers six real-world NLP tasks and six dimensions of commonsense reasoning types.
3. The constructed method to generate the commonsense reasoning tasks is make sense. Meanwhile, the authors also consider the fine-grained dimensions for further analysis.
4. The baseline models in experiments are sufficient.


**Reasons To Reject:**

1. One major concern with the paper is the lack of clarity in its structure and statements, which makes it challenging to follow the content. For instance, Figures 1 and 2 depict the CRoW Evaluation as a multi-choice commonsense reasoning task, but the paper states that "all tasks in CRoW are treated as binary classification tasks." Improving the clarity of these statements would significantly enhance the overall readability and understanding of the paper.
2. The setup of only two annotators in the human evaluation raises concerns about the rigor and reliability of the evaluation process. In general, human evaluations should involve more than three annotators to ensure more robust and dependable results.


**Reproducibility:**

4: Could mostly reproduce the results, but there may be some variation because of sample variance or minor variations in their interpretation of the protocol or method.

**Reviewer Confidence:**

4: Quite sure. I tried to check the important points carefully. It's unlikely, though conceivable, that I missed something that should affect my ratings.

---

> ### Author Rebuttal · Authors · 2023-08-29
>
> We appreciate the reviewer’s comments and suggestions. We are happy to see that the reviewer is excited about our work.
>
> **Discrepancy between intro Figure and the experimental setup in terms of task setting**
>
> We propose to evaluate the commonsense reasoning skills in a binary classification setting and report macro-F1 and situation accuracy (for measuring robustness). In our intro Figure, we intended to demonstrate the contrast between different negative samples included in examples of our benchmark. We agree that this does give the false appearance of a multi-choice task and will fix this misunderstanding in the camera-ready version.
>
> **Why were there only 2 annotators chosen for the human evaluation?**
>
> We thank the reviewer for bringing up these important points about the reliability of our human evaluation results. First, our setup may only have two annotators per task, but these are expert annotators who are NLP researchers from our lab (importantly, they were not involved in the original data collection). As a result, the evaluators are more experienced, can clarify misunderstandings in the annotation guideline with us, and generally produce more careful annotations than crowd workers. Finally, following [1, 2], we intentionally allowed annotators to discuss in cases of disagreement to reduce variance and yield a robust upper bound for our task. In the camera-ready version, we will clarify these important points about our human study.
>
> [1] Amidei, Jacopo et al. 2018 Rethinking the Agreement in Human Evaluation Tasks, ACL
>
> [2] Oortwijn et al., 2021 Interrater Disagreement Resolution: A Systematic Procedure to Reach Consensus in Annotation Tasks, ACL

---

### Official Review · Reviewer_1q4f · 2023-08-12

**Soundness:** 3

**Excitement:**

4: Strong: This paper deepens the understanding of some phenomenon or lowers the barriers to an existing research direction.

**Paper Topic And Main Contributions:**

Proposes a way to construct a commonsense reasoning dataset that is more grounded in 'real-world' scenarios. It crowdsources annotations of commonsense knowledge using a NLP task to provide examples of a context string and a target string. After commonsense knowledge validation, crowdsources the perturbation of the target to generate a Winograd schema.

It contributes CRoW, a commonsense reasoning dataset consisting of 6 NLP tasks (Machine translation, Open-domain Dialogue, Dialogue summarization, Intent detection, Stance classification, Safety detection) and 6 types of 'commonsense knowledge dimensions' (Attributional, Physical/Spatial, Temporal, Causal, Social and Comparative). It evaluates several LLMs on the constructed dataset to set benchmarks on the 6 NLP tasks as well as on the 6 commonsense knowledge dimensions.

**Questions For The Authors:**

Question A: In line 320-328, was there any ranking among the 25 potential tasks? How were the tasks chosen to be part of CRoW? Were intent detection, stance classification, and safety detection part of these 25?

Question B: In line 261-268, to confirm, this conveys that other approaches ignore (3) and (4) while your technique does not?

Question C: In Figure 4, what does the image convey?
In the first example, everything except GPT-4 fails? What about GPT-4 with CoT or oracle knowledge?
In the second example, everything except GPT-4 + CoT fails? While CoT generally improves the result, you have provided reasoning and analysis that GPT-4+CoT performs worse that base GPT-4.

Question D: In Table 4, there are 2 values per column. Are they Macro F1 and Situational Accuracy? The oracle knowledge that is added is only for negative examples? Can you please expand more on footnote 4? The negative/wrong knowledge triplet would be rejected in stage 1, CKV?

Question E: Can you explain why only 2 human annotators were chosen for the human benchmark? 2 is not enough to get general consensus and majority voting. The part about human discussion in case of no majority shows that it may not rigorous and proper.

**Reasons To Accept:**

The framework for constructing a commonsense reasoning dataset grounded in a real world NLP context seems like a fairly innovative, albeit minor, modification of existing techniques. It does seem to make the resulting dataset more nuanced to test the understanding of current and future models as the authors have shown a decent gap in performance between the humans and the LLMs.

The CRoW dataset is quite diverse and spans a wide variety of NLP tasks and different types of commonsense knowledge. The annotation guidelines are clear and the dataset quality verification is good. The evaluation of different LLMs on the dataset are interesting and it shows a decent gap between the human benchmark and the model performances.

**Reasons To Reject:**

The human annotation size was limited to get a true estimate of human performance. Having only 2 annotators draws questions about reliability of this metric and tie break in case of no majority vote. Having discussion between the annotators to come to a conclusion makes the process not have independent annotators which means this was not robust.

The task selection part needed some justification on the tasks chosen to be included in the construction of CRoW. Same with the commonsense knowledge dimension selection.

The performance analysis of the LLMs on the different dimensions of commonsense knowledge was slightly lacking. This showed that all the different dimensions are nearly the same with not much distinction between them.

The analysis and reasoning for the performance difference between GPT4 and GPT4+CoT was relatively weak compared to the rest of the analysis.

The methodology of only using negative samples from the CKA stage for evaluating oracle knowledge with the LLMs may skew the results as only one part of the LLM prediction change will be captured.

**Reproducibility:**

N/A: Doesn't apply, since the paper does not include empirical results.

**Reviewer Confidence:**

3: Pretty sure, but there's a chance I missed something. Although I have a good feel for this area in general, I did not carefully check the paper's details, e.g., the math, experimental design, or novelty.

**Typos Grammar Style And Presentation Improvements:**

Suggestion A: For lines 181-186, please rephrase as it was a bit confusing.

Suggestion B: In figure 2, show more than 1 example in WSG of false intent (similar to CKA)

Suggestion C: In Figure 3, category 'other' can not be seen clearly. Maybe change the colors or add a percentage in the graphs.

Suggestion D: In Lines 261-268 give examples for each condition, 1 positive and 1 negative

Suggestion E: Figure 4 caption to be expanded. Maybe add one failure and one success response at each subfigure. Also clearly mention what all failed and what succeeded.

Suggestion F: Table 4 caption to be expanded. Maybe table 3&4 can be expanded to include more models. Add param numbers from table 2 to maintain consistency with table 3&4.

---

> ### Author Rebuttal · Authors · 2023-08-29
>
> We thank the reviewer for their thoughtful comments and helpful suggestions. We accept and intend to incorporate their suggestions in the camera-ready version of our paper.
>
> **Why were there only 2 annotators chosen for the human evaluation and why was there a discussion between annotators to resolve disagreements?**
>
> We thank the reviewer for bringing up these important points about the reliability of our human evaluation results. However, we actually think these points stem from the potential advantages of our study. First, our setup may only have two annotators per task, but these are expert annotators who are NLP researchers from our lab (importantly, they were not involved in the original data collection). As a result, the evaluators are more experienced, can clarify misunderstandings in the annotation guideline with us, and generally produce more careful annotations than crowd workers. Finally, following [1, 2], we intentionally allowed annotators to discuss in cases of disagreement to reduce variance and yield a robust upper bound for our task. In the camera-ready version, we will clarify these important points about our human study.
>
> [1] Amidei, Jacopo et al. 2018 Rethinking the Agreement in Human Evaluation Tasks, ACL
>
> [2] Oortwijn et al., 2021 Interrater Disagreement Resolution: A Systematic Procedure to Reach Consensus in Annotation Tasks, ACL
>
> **How did we select the NLP tasks and was there any ranking used in the selection?**
>
> As mentioned in section 4 of our paper, following (Davis, 2023), we identify the following desiderata of tasks in the benchmark: (1) tasks should have real-world applications (e.g., machine translation) and (2) tasks should involve rich commonsense knowledge and reasoning. We choose these criteria since our benchmark targets the evaluation of commonsense knowledge and reasoning in practical NLP applications.
> To choose NLP tasks, we first crawl papers from the whole ACL anthology published since year 2000 (approximated 94K papers). Then, we automatically select those papers that have performed some error analysis and mention commonsense or world knowledge in their categories of errors. These steps resulted in 25 potential tasks.
> Out of these 25 tasks, we selected tasks based on the difficulty of generating commonsense violating perturbations. After manually designing commonsense perturbations for all 25 tasks that we found in our literature review, we select 6 tasks for which we could design a scalable pipeline for generating perturbations that could be reliably completed by crowdworkers.
>
>
> **How did we select the Commonsense Knowledge Dimensions?**
>
> We considered widely used commonsense knowledge bases such as ConceptNet (Speer et al., 2016) and ATOMIC (Sap et al., 2019a) as well as recent works such as ATOMIC2020 (Hwang et al., 2020) and CIDER (Ghosal et al., 2021) for selecting the commonsense relations. As an initial step, the authors’ manually categorized the kind of knowledge relations that appeared for each task. Among the total 56 relations available to us from these sources, we found 22 relations from ConceptNet (out of 36), 8 relations from ATOMIC (out of 9), 3 relations from ATOMIC2020 (out of 5) and 3 relations from CIDER (out of 6) appearing most commonly. Following Ghosal et al., 2021 and Ilievski et al., 2021, we further categorized these 36 relations into 6 commonsense knowledge dimensions.
>
> **Do previous works ignore the following conditions while designing Winograd schemas: (3) the target must be an incorrect answer for the given context, and (4) the target is contextually relevant?**
>
> We note that while other approaches do not explicitly ignore these conditions, they also do not explicitly address them. Hence, in our work we explicitly ask our annotators to consider these conditions to ensure the high difficulty of generations in our benchmark.
>
> **Limited CK dimension analysis**
>
> The reviewer finds the overall evaluation results interesting, but that the analysis on the different dimensions of commonsense knowledge shows performance on most dimensions is nearly the same with not much distinction between them.
> We agree with R1 that while we report a significant gap in overall evaluation of LLMs on our benchmark, our analysis on the performance of these models across different dimensions does not show a significant difference in how the model performs on different types of knowledge. However, we believe this sheds light on how language models represent commonsense knowledge, namely, with inherent interconnectivity across dimensions. More specifically, commonsense knowledge dimensions and relations are not mutually exclusive and one piece of knowledge typically involves several dimensions, while, one dimension is naturally more salient and that knowledge hence is represented by a particular relation in that dimension. Therefore, mastering a particular dimension of knowledge by a language model might in fact, amount to mastering several or all dimensions. Hence, we see uniformity in the performance of these models across dimensions.
>
> **In Figure 4, what does the image convey? In the first example, everything except GPT-4 fails? What about GPT-4 with CoT or oracle knowledge? In the second example, everything except GPT-4 + CoT fails? While CoT generally improves the result, you have provided reasoning and analysis that GPT-4+CoT performs worse that base GPT-4.**
>
> Figure 4 was designed to showcase examples from a few contrastive scenarios (small vs large models, simple prompting vs chain-of-thought prompting etc.), therefore was not meant to convey a comprehensive view of all the possible failure cases as there are too many. However, we agree that the figure could be clearer and we will add these details for better clarity in the camera-ready version. More specifically, in the first example, we meant all models that are not GPT-4, hence GPT-4 with oracle knowledge or CoT were not considered. In the second example, we specifically look at the effect of CoT, so only GPT-4 is considered and we include an example where GPT-4 with CoT does improve the result. We realize that including an example where it does not improve would be better as it is also aligned with our findings and we intend to add them in the final version.
>
> **Analysis and reasoning for the performance difference between GPT-4 and GPT-4+CoT**
>
> The reviewer finds the analysis and reasoning for the performance difference between GPT-4 and GPT-4+CoT weak. In this answer, we would like to expand on our analysis and reasoning with an example and we intend to incorporate these additional points in the camera-ready version. We believe the worse performance of GPT-4 with CoT prompting compared to GPT-4 with simple prompting is due to the GPT-4’s more imaginative, but less plausible reasoning process triggered by the CoT prompting. As an example, we can look at the following dialogue, response pair and GPT-4’s answer on whether it is plausible when prompted with CoT.
> ```
> Dialogue:
>             A: Are you ready to move?
>             B: Definitely. I Ve got the hell out of the company. Anything new?
>             A: Yes, but there is a chance that might be suitable for you.
>             B: Really? Are you willing to recommend me for the position?
>             A: I can't say, but let me make a call and see if I can help.
> Response:
> B: I appreciate the lack of support, sir.
> Step 1: Understand the dialogue
> Person A asks if Person B is ready to move. Person B confirms and asks if there is anything new. Person A mentions a possible opportunity but is unsure if they can recommend Person B. Person A offers to make a call to see if they can help.
> Step 2: Analyze the response
> The response from Person B seems to be sarcastic, as they are expressing appreciation for the "lack of support" when Person A has actually offered to make a call to help.
> Step 3: Determine if the response can plausibly follow the dialogue
> The response can plausibly follow the dialogue, as Person B might be sarcastically expressing their feelings about the situation.
> Final answer: <Answer>Yes</Answer>
> ```
> When we analyze the error made by GPT-4 with CoT, we see that GPT-4 thought the sarcastic response was plausible. While this response could technically be possible, it might also violate our commonsense idea of what would be a reasonable response to a helpful query. Overall, we find other such examples where GPT-4 w/ CoT underperforms GPT-4 due to sarcastic or metaphorical elements in this dialogue.
>
> **In Table 4, there are 2 values per column. Are they Macro F1 and Situational Accuracy?**
>
> Thank you for pointing it out, the results reported in Table 4 correspond to Macro-F1 and Situational Accuracy and we will fix the caption to reflect this.
>
> **Would wrong knowledge triplet be rejected in stage 1, CKV? Distinction between negative sample and wrong knowledge triplet?**
>
> First, we would like to clarify the distinction between the negative sample and negative (wrong) knowledge. Negative (wrong) knowledge is filtered out in our CKV stage. Negative samples do not refer to this as knowledge, but instead the constructed Winograd schema that are “semantically incorrect” based on a piece of commonsense knowledge. Second, the annotated commonsense knowledge is about both negative and positive samples, since this knowledge connects the context with the positive target and is used to generate the negative target.
>
> **Why did we use only negative samples for evaluating oracle knowledge with LLMs?**
>
> During our annotation pipeline, commonsense dimensions are linked explicitly to the negative samples (which are constructed using a provided commonsense violation). For positive examples, multiple pieces of commonsense may be annotated (from the CKA stage). As a result, there is no 1-1 correspondence between positive samples and a piece of commonsense knowledge (i.e., no clear “oracle” knowledge”), so we opted to analyze the performance of these models on the commonsense knowledge dimension for negative samples only and report the results in Table 4.

---

### Meta-Review · Area_Chair_tP3d · 2023-09-18

**Recommendation:** 2

**Metareview:**

he paper presents CROW, a benchmark for evaluating commonsense reasoning in real-world NLP tasks. All reviewers viewed the framework for constructing a commonsense reasoning dataset grounded in a real world NLP context as fairly innovative. However, the reviewers also raised some concerns about the soundness (details) of the construction procedure of the dataset.  Most of these concerns were at least partially addressed by the authors during the rebuttal phase. I recommend the authors to add more analysis of stableness.

---

### Decision · Program_Chairs · 2023-10-07

**Decision:**

Accept-Main

**Comment:**

he paper presents CROW, a benchmark for evaluating commonsense reasoning in real-world NLP tasks. All reviewers viewed the framework for constructing a commonsense reasoning dataset grounded in a real world NLP context as fairly innovative. However, the reviewers also raised some concerns about the soundness (details) of the construction procedure of the dataset.  Most of these concerns were at least partially addressed by the authors during the rebuttal phase. I recommend the authors to add more analysis of stableness.